# ARE GENERATIVE CLASSIFIERS MORE ROBUST TO ADVERSARIAL ATTACKS?

## ABSTRACT

There is a rising interest in studying the robustness of deep neural network classifiers against adversaries, with both advanced attack and defence techniques being actively developed. However, most recent work focuses on *discriminative* classifiers, which only model the conditional distribution of the labels given the inputs. In this paper, we propose and investigate the *deep Bayes* classifier, which improves classical naive Bayes with conditional deep generative models. We further develop detection methods for adversarial examples, which reject inputs with low likelihood under the generative model. Experimental results suggest that deep Bayes classifiers are more robust than deep discriminative classifiers, and that the proposed detection methods are effective against many recently proposed attacks.

## 1 INTRODUCTION

Deep neural networks have been shown to be vulnerable to adversarial examples (Szegedy et al., 2013; Goodfellow et al., 2014). The latest attack techniques can easily fool a deep neural network with imperceptible perturbations (Goodfellow et al., 2014; Papernot et al., 2016b; Carlini & Wagner, 2017a; Kurakin et al., 2016; Madry et al., 2018; Chen et al., 2017a), even in the black-box case, where the attacker does not have access to the network's weights (Papernot et al., 2017b; Chen et al., 2017b; Alzantot et al., 2018a). Adversarial attacks are serious security threats to machine learning systems, threatening applications beyond image classification (Carlini & Wagner, 2018; Alzantot et al., 2018b).

To address this outstanding security issue, researchers have proposed defence mechanisms against adversarial attacks. Adversarial training, which augments the training data with adversarially perturbed inputs, has shown moderate success at defending against recently proposed attack techniques (Szegedy et al., 2013; Goodfellow et al., 2014; Tramèr et al., 2017; Madry et al., 2018). In addition, recent advances in Bayesian neural networks have demonstrated that uncertainty estimates can be used to detect adversarial attacks (Li & Gal, 2017; Feinman et al., 2017; Louizos & Welling, 2017; Smith & Gal, 2018). Another notable category of defence techniques involves the usage of generative models. For example, Gu & Rigazio (2014) used an auto-encoder to denoise the inputs before feeding them to the classifier. This denoising approach has been extensively investigated, and the "denoisers" in usage include generative adversarial networks (Samangouei et al., 2018), PixelCNNs (Song et al., 2018) and denoising auto-encoders (Kurakin et al., 2018). These developments rely on the *"off-manifold"* conjecture, that is, that adversarial examples are far away from the data manifold, although Gilmer et al. (2018) has challenged this observation with a synthetic example.

Surprisingly, much less recent work has investigated the robustness of *generative classifiers* (Ng & Jordan, 2002) against adversarial attacks for multi-class classification, where such classifiers explicitly model the conditional distribution of the inputs given labels. Typically, a generative classifier produces predictions by comparing between the likelihood of the labels for a given input, which is closely related to the "distance" of the input to the data manifold associated with a class. Therefore, generative classifiers should be robust to many recently proposed adversarial attacks if the "off-manifold" conjecture holds for many real-world applications. Unfortunately, many generative classifiers in popular use, including naive Bayes and linear discriminant analysis (Fisher, 1936), perform poorly on natural image classification tasks, making it difficult to verify the "off-manifold" conjecture and the robustness of generative classifiers with these tools. In recent work, k-nearest neighbors (Cover & Hart, 1967), a method which shares many similarities with generative classi-

fiers, has been significantly improved in handling natural images by leveraging deep feature representations (Papernot & McDaniel, 2018). To the best of our knowledge, an approach which targets a similar contribution has not yet been proposed for generative classifiers.

Are generative classifiers more robust to recently proposed adversarial attack techniques? To answer this, we improve the naive Bayes algorithm by using conditional deep generative models, and evaluate the conjecture on the proposed generative classifier. In summary, our contributions include:

- We propose *deep Bayes* as an extension of naive Bayes, in which the conditional distribution of an input, given a label, is parameterised by a deep latent variable model (LVM). We learn the LVM with the variational auto-encoder algorithm (Kingma & Welling, 2013; Rezende et al., 2014), and approximate Bayes' rule using importance sampling.
- We propose three detection methods for adversarial perturbations. The first two use the learned generative model as a proxy of the data manifold, and reject inputs that are far away from it. The third computes statistics for the classifier's output probability vector on training data, and rejects inputs that lead to under-confident predictions.
- We evaluate the robustness of the proposed generative classifier on the MNIST multi-class and CIFAR binary classification tasks. We also improve the robustness of deep neural networks on CIFAR-10 multi-class classification, by fusing discriminatively learned visual feature representations with the proposed generative classifiers. We further compare the generative classifiers with a number of discriminative classifiers, including Bayesian neural networks and *discriminative* latent variable models.

## 2 DEEP BAYES: CONDITIONAL DEEP LVM AS A GENERATIVE CLASSIFIER

Denote $p_{\mathcal{D}}(\boldsymbol{x}, \boldsymbol{y})$ the data distribution for the input $\boldsymbol{x} \in \mathbb{R}^D$ and label $\boldsymbol{y} \in \{\boldsymbol{y}_c | c = 1, ..., C\}$, where $\boldsymbol{y}_c$ denotes the one-hot encoding vector for class $c$. For a given $\boldsymbol{x} \in \mathbb{R}^D$ we can define the ground-truth label by

$$\boldsymbol{y} \sim p_{\mathcal{D}}(\boldsymbol{y}|\boldsymbol{x}) \quad \text{if } \boldsymbol{x} \in \text{supp}(p_{\mathcal{D}}(\boldsymbol{x})). \tag{1}$$

We assume the data distribution $p_{\mathcal{D}}(\boldsymbol{x}, \boldsymbol{y})$ follows the *manifold assumption*: for every class $c$, the conditional distribution $p_{\mathcal{D}}(\boldsymbol{x}|\boldsymbol{y}_c)$ has a low-dimensional manifold support $\mathcal{M}_c = \text{supp}(p_{\mathcal{D}}(\boldsymbol{x}|\boldsymbol{y}_c))$. Therefore the training dataset $\mathcal{D} = \{(\boldsymbol{x}^{(n)}, \boldsymbol{y}^{(n)})\}_{n=1}^N$ is generated as the following:

$$(\boldsymbol{x}^{(n)}, \boldsymbol{y}^{(n)}) \sim p_{\mathcal{D}}(\boldsymbol{x}, \boldsymbol{y}) \iff \boldsymbol{y}^{(n)} \sim p_{\mathcal{D}}(\boldsymbol{y}), \boldsymbol{x}^{(n)} \sim p_{\mathcal{D}}(\boldsymbol{x}|\boldsymbol{y}).$$

A (Bayesian) generative classifier first builds a *generative model* $p(\boldsymbol{x}, \boldsymbol{y}) = p(\boldsymbol{x}|\boldsymbol{y})p(\boldsymbol{y})$, and then, in prediction time, predicts the label $\boldsymbol{y}^*$ of a test input $\boldsymbol{x}^*$ using Bayes' rule,

$$p(\boldsymbol{y}^*|\boldsymbol{x}^*) = \frac{p(\boldsymbol{x}^*|\boldsymbol{y}^*)p(\boldsymbol{y}^*)}{p(\boldsymbol{x}^*)} = \text{softmax}_{c=1}^C \left[\log p(\boldsymbol{x}^*, \boldsymbol{y}_c)\right],$$

where $\text{softmax}_{c=1}^C$ denotes the softmax operator over the $c$ axis. Therefore, the output probability vector is computed analogously to many deep discriminative classifiers which use softmax activation in the output layer, so many existing attacks can be tested directly. However, unlike discriminative classifiers, the "logit" values prior to softmax activation have a clear meaning here, which is the (approximated) log joint distribution $\log p(\boldsymbol{x}^*, \boldsymbol{y}_c)$ of input $\boldsymbol{x}^*$ conditioned on a given label $\boldsymbol{y}_c$. Therefore, one can also analyse the logit values to determine whether the unseen pair $(\boldsymbol{x}^*, \boldsymbol{y}^*)$ is legitimate, a utility which will be discussed further in later sections.

*Naive Bayes* is perhaps the most well-known generative classifier; it assumes a factorised distribution for the conditional generator, i.e. $p(\boldsymbol{x}|\boldsymbol{y}) = \prod_{d=1}^D p(x_d|\boldsymbol{y})$. However, this factorisation assumption is inappropriate for e.g. image and speech data. Fortunately, we can leverage the recent advances in generative modelling and apply a deep generative model for the joint distribution $p(\boldsymbol{x}, \boldsymbol{y})$. We refer to such generative classifiers that use deep generative models as *deep Bayes* classifiers.

In this paper, we consider a deep latent variable model (LVM) $p(\boldsymbol{x}, \boldsymbol{z}, \boldsymbol{y})$, which will be used for classification. In this case, the conditional distribution is $p(\boldsymbol{x}|\boldsymbol{y}) = \frac{\int p(\boldsymbol{x}, \boldsymbol{z}, \boldsymbol{y})d\boldsymbol{z}}{\int p(\boldsymbol{x}, \boldsymbol{z}, \boldsymbol{y})d\boldsymbol{z}d\boldsymbol{x}}$. Importantly, this leads to a conditional distribution $p(\boldsymbol{x}|\boldsymbol{y})$ that is *not* factorised (even when $p(\boldsymbol{x}|\boldsymbol{z}, \boldsymbol{y})$ is), which is much more powerful than naive Bayes. Since maximum likelihood is intractable, we follow

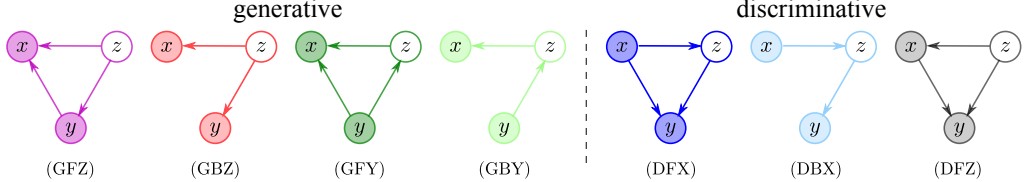

Figure 1: A visualisation of the graphical models, including both **G**enerative and **D**iscriminative ones, as well as **F**ully connected and **B**ottleneck ones. The last character indicates the first node in the topological order of the graph. The colour encoding is the same as those in experiments.

Kingma & Welling (2013) and Rezende et al. (2014) to introduce an amortised approximate posterior $q(\boldsymbol{z}|\boldsymbol{x},\boldsymbol{y})$, and train both $p$ and $q$ by maximising the variational lower-bound:

$$\mathbb{E}_{\mathcal{D}}[\mathcal{L}_{\text{VI}}(\boldsymbol{x},\boldsymbol{y})] = \frac{1}{N}\sum_{n=1}^{N}\mathbb{E}_q\left[\log\frac{p(\boldsymbol{x}_n,\boldsymbol{z}_n,\boldsymbol{y}_n)}{q(\boldsymbol{z}_n|\boldsymbol{x}_n,\boldsymbol{y}_n)}\right]. \tag{2}$$

After training, the predicted class probability vector $\boldsymbol{y}^*$ for a future input $\boldsymbol{x}^*$ is computed by an approximation to Bayes' rule with importance sampling:

$$p(\boldsymbol{y}^*|\boldsymbol{x}^*) \approx \text{softmax}_{c=1}^{C}\left[\log\frac{1}{K}\sum_{k=1}^{K}\frac{p(\boldsymbol{x}^*,\boldsymbol{z}_c^k,\boldsymbol{y}_c)}{q(\boldsymbol{z}_c^k|\boldsymbol{x}^*,\boldsymbol{y}_c)}\right], \quad \boldsymbol{z}_c^k \sim q(\boldsymbol{z}|\boldsymbol{x}^*,\boldsymbol{y}_c). \tag{3}$$

We evaluate the effect of the following factorisation structures on the robustness of the induced classifier from the generative model $p(\boldsymbol{x},\boldsymbol{z},\boldsymbol{y})$ (see Figure 1).

$p(\boldsymbol{x},\boldsymbol{z},\boldsymbol{y}) = p(\boldsymbol{z})p(\boldsymbol{y}|\boldsymbol{z})p(\boldsymbol{x}|\boldsymbol{z},\boldsymbol{y})$     (GFZ)

$p(\boldsymbol{x},\boldsymbol{z},\boldsymbol{y}) = p_{\mathcal{D}}(\boldsymbol{y})p(\boldsymbol{z}|\boldsymbol{y})p(\boldsymbol{x}|\boldsymbol{z},\boldsymbol{y})$     (GFY)

$p(\boldsymbol{x},\boldsymbol{z},\boldsymbol{y}) = p(\boldsymbol{z})p(\boldsymbol{y}|\boldsymbol{z})p(\boldsymbol{x}|\boldsymbol{z})$     (GBZ)

$p(\boldsymbol{x},\boldsymbol{z},\boldsymbol{y}) = p_{\mathcal{D}}(\boldsymbol{y})p(\boldsymbol{z}|\boldsymbol{y})p(\boldsymbol{x}|\boldsymbol{z})$     (GBY)

$p(\boldsymbol{x},\boldsymbol{z},\boldsymbol{y}) = p_{\mathcal{D}}(\boldsymbol{x})p(\boldsymbol{z}|\boldsymbol{x})p(\boldsymbol{y}|\boldsymbol{z},\boldsymbol{x})$     (DFX)

$p(\boldsymbol{x},\boldsymbol{z},\boldsymbol{y}) = p(\boldsymbol{z})p(\boldsymbol{x}|\boldsymbol{z})p(\boldsymbol{y}|\boldsymbol{z},\boldsymbol{x})$     (DFZ)

$p(\boldsymbol{x},\boldsymbol{z},\boldsymbol{y}) = p_{\mathcal{D}}(\boldsymbol{x})p(\boldsymbol{z}|\boldsymbol{x})p(\boldsymbol{y}|\boldsymbol{z})$     (DBX)

We use the initial character "G" to denote generative classifiers and "D" to denote discriminative classifiers. Models with the second character as "F" have a *fully connected* graph, while "B" models enforce the usage of the latent code $\boldsymbol{z}$ as a *bottleneck*. The last character of the model name indicates the first node in topological order. Model DFZ is somewhat intermediate, as it builds a generative model for the inputs $\boldsymbol{x}$ but also directly parameterises the conditional distribution $p(\boldsymbol{y}|\boldsymbol{x},\boldsymbol{z})$. We do not test other architectures under this nomenclature, as either the graph contains directed cycles (e.g. $\boldsymbol{x} \rightarrow \boldsymbol{y} \rightarrow \boldsymbol{z} \rightarrow \boldsymbol{x}$), or $\boldsymbol{z}$ is the last node in topological order (e.g. $\boldsymbol{x} \rightarrow \boldsymbol{y}, (\boldsymbol{x},\boldsymbol{y}) \rightarrow \boldsymbol{z}$) so that the marginalisation of $\boldsymbol{z}$ does not affect classification.

## 3   DETECTING ADVERSARIAL ATTACKS WITH GENERATIVE CLASSIFIERS

We propose detection methods for adversarial examples using generative classifier's logit values. As an illustrating example, consider a labelled dataset of "cat" and "dog" images. If an adversarial image of a cat $\boldsymbol{x}_{\text{adv}}$ is incorrectly labelled as "dog", then either this image is ambiguous, or, under a *perfect* generative model, the logit $\log p(\boldsymbol{x}_{\text{adv}}, \text{"dog"})$ will be significantly lower than normal. This means we can detect attacks using the logits $\log p(\boldsymbol{x}^*,\boldsymbol{y}_c), c = 1,...,C$ computed on a test input $\boldsymbol{x}^*$, by comparing them with the logits computed on legitimate training inputs.

Concretely, the proposed detection algorithms are as follows. We aim to reject both unlabelled input $\boldsymbol{x}$ that have low probability under $p_{\mathcal{D}}(\boldsymbol{x})$, and labelled data $(\boldsymbol{x},\boldsymbol{y})$ that have low $p_{\mathcal{D}}(\boldsymbol{x},\boldsymbol{y})$ values.

- **Marginal detection**: rejecting inputs that are far away from the manifold.
  One can select a threshold $\delta$ and reject an input $\boldsymbol{x}$ if $-\log p(\boldsymbol{x}) > \delta$. To determine the threshold $\delta$, we can compute $\bar{d}_{\mathcal{D}} = \mathbb{E}_{\boldsymbol{x}\sim\mathcal{D}}[-\log p(\boldsymbol{x})]$ and $\sigma_{\mathcal{D}} = \sqrt{\mathbb{V}_{\boldsymbol{x}\sim\mathcal{D}}[\log p(\boldsymbol{x})]}$, then set $\delta = \bar{d}_{\mathcal{D}} + \alpha\sigma_{\mathcal{D}}$. It is also possible to compute the statistics $\bar{d}_p, \sigma_p$ on the images generated by the generative model accordingly.
- **Logit detection**: rejecting inputs using joint density.
  Given a victim model $\boldsymbol{y} = F(\boldsymbol{x})$, one can reject $\boldsymbol{x}$ if $-\log p(\boldsymbol{x}, F(\boldsymbol{x})) > \delta_{\boldsymbol{y}}$. We can use the mean and variance statistics $\bar{d}_c, \sigma_c$ computed on $\log p(\boldsymbol{x},\boldsymbol{y}_c)$ and select $\delta_{\boldsymbol{y}_c} = \bar{d}_c + \alpha\sigma_c$.

Figure 2: Visualising detection mechanisms. The scattered dots are training data points, with different classes shown in different colours (red for $c = 0$ and blue for $c = 1$). Same labels are manually assigned for inputs when the detection method requires $\boldsymbol{y}$. Decision regions are shown in the corresponding colours. Input points in the shaded area are rejected by detection.

- **Divergence detection**: rejecting inputs with over- and/or under-confident predictions. Denote $\boldsymbol{p}(\boldsymbol{x})$ as a $C$-dimensional probability vector outputted by the classifier. For each class $c$, we first collect the *mean classification probability vector* $\boldsymbol{p}_c = \mathbb{E}_{(\boldsymbol{x},\boldsymbol{y}_c)\in\mathcal{D}}[\boldsymbol{p}(\boldsymbol{x})]$, then compute the mean $\bar{d}_c$ and standard deviation $\sigma_c$ on a selected divergence/distance measure $\mathrm{D}[\boldsymbol{p}_c||\boldsymbol{p}(\boldsymbol{x})]$ for all $(\boldsymbol{x}, \boldsymbol{y}_c) \in \mathcal{D}$. A test input $\boldsymbol{x}^*$ with prediction label $c^* = \arg\max \boldsymbol{p}(\boldsymbol{x}^*)$ is rejected if $\mathrm{D}[\boldsymbol{p}_{c^*}||\boldsymbol{p}(\boldsymbol{x}^*)] > \bar{d}_{c^*} + \alpha\sigma_{c^*}$. Therefore, an example $\boldsymbol{x}^*$ will be rejected if the classifier is over-confident or under-confident (ambiguous inputs).
  When D is selected as the KL-divergence, we call this detection method *KL detection*. Other divergence/distance measures such as total variation (TV) can also be used.

For better intuition, we visualise the detection mechanisms in Figure 2 with a synthetic "two rings" binary classification example. In this case we sample $2 \times 1000$ training data points as:

$$(\boldsymbol{x}, \boldsymbol{y}) \sim p_{\mathcal{D}} \Leftrightarrow \boldsymbol{y} \sim \mathrm{Bern}(0.5), \theta \sim \mathrm{Uniform}(0, 2\pi), \boldsymbol{x}|\boldsymbol{y} \sim \mathcal{N}(\boldsymbol{x}; \boldsymbol{c}_{\boldsymbol{y}} + r_{\boldsymbol{y}}[cos(\theta), sin(\theta)]^{\mathrm{T}}, \sigma^2\mathbf{I}).$$

We consider a generative classifier $p(\boldsymbol{x}, \boldsymbol{y}) = p(\boldsymbol{x}|\boldsymbol{y})p_{\mathcal{D}}(\boldsymbol{y}) = \mathcal{N}(\boldsymbol{x}; \boldsymbol{\mu}_{\boldsymbol{y}}, \sigma^2\mathbf{I})\mathrm{Bern}(0.5)$ with $\boldsymbol{\mu}_{\boldsymbol{y}} = \mathrm{proj}(\boldsymbol{x}, \mathrm{ring}_{\boldsymbol{y}}) = \arg\min_{\hat{\boldsymbol{x}}\in\mathbb{R}^2, ||\hat{\boldsymbol{x}}-\boldsymbol{c}_{\boldsymbol{y}}||_2=r_{\boldsymbol{y}}} ||\boldsymbol{x} - \hat{\boldsymbol{x}}||_2$. The $\delta$ thresholds are selected to achieve 10% false positive rates on training data. From the visualisations we see that inputs that are far away from the model manifold are rejected by marginal/logit detection. At the same time, logit detection rejects data points that are not on the manifold of the given class. KL/TV detection does not construct manifold-aware acceptance regions, which is as expected since the proposed divergence detection method does not require the classifier to be generative. However, both detection methods have some success in rejecting uncertain predictions, especially for TV, which also rejects ambiguous inputs (see the ring-cross regions in the last two plots). Combining all three methods, we see that the rejected inputs are either far away from the manifold, or are ambiguous. Again we emphasise that a suitable generative model is required to make the detection methods work in practice, since the data distribution $p_{\mathcal{D}}(\boldsymbol{x}, \boldsymbol{y})$ is approximated by $p(\boldsymbol{x}, \boldsymbol{y})$.

Detection methods using logit values have also been proposed in Li & Gal (e.g. 2017); Feinman et al. (e.g. 2017). However it is unclear whether the logits values in discriminative classifiers have a clear semantic meaning, while the logit values in deep Bayes represent the log probability of generating the input $\boldsymbol{x}$ given the class label $\boldsymbol{y} = \boldsymbol{y}_c$. Our approach is also distinct from previous "denoising" approaches (e.g. Song et al., 2018; Samangouei et al., 2018; Kurakin et al., 2018) that require training an additional generative model separately. The deep Bayes classifiers share the same generative model as the detection methods, meaning that detected adversarial examples are indeed far away from the classifier's manifold (which is also an approximation to the data manifold). Therefore the claim "detection neural networks can be bypassed" (Carlini & Wagner, 2017a; Athalye et al., 2018) does not directly transfer to our approach, and we can use the marginal and logit detection methods to directly verify the "off-manifold" adversarial example conjecture.

# 4 EXPERIMENTS

We carry out a number of tests on the proposed deep Bayes classifiers (3), where $q(\boldsymbol{z}|\cdot)$ and $p(\boldsymbol{z}|\cdot)$ are factorised Gaussians, and the conditional probability $p(\boldsymbol{x}|\cdot)$, if required, is parameterised by an $\ell_2$ loss. Besides the LVM-based classifiers, we further train a Bayesian neural network (BNN) with Bernoulli dropout (dropout rate 0.3), as it has been shown in Li & Gal (2017) and Feinman et al. (2017) that BNNs are more robust than their deterministic counterparts. The constructed BNN has 2x more channels than LVM-based classifiers, making the comparison slightly "unfair", as the BNN layers have more capacity. We use $K = 10$ Monte Carlo samples for all the classifiers.

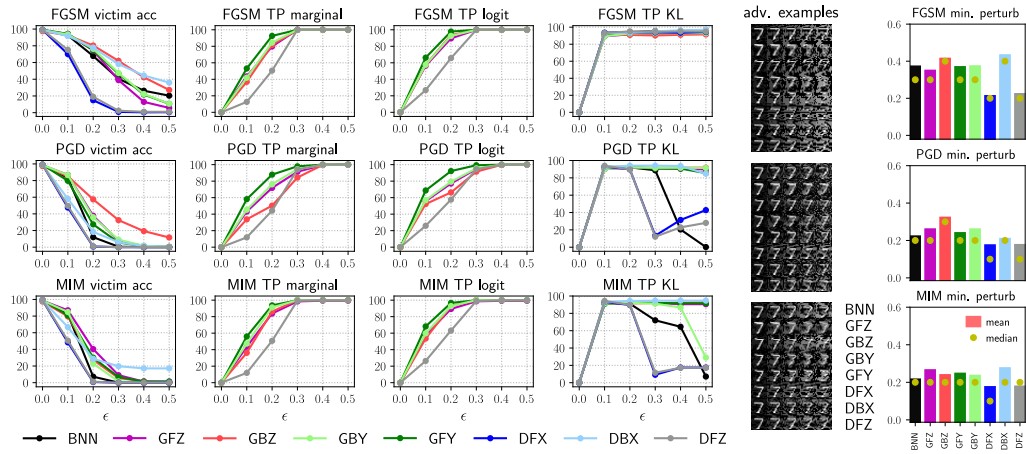

Figure 3: Accuracies (column 1), detection rates (columns 2-4) and minimum $\ell_{\inf}$ perturbation (column 6) against **white-box zero-knowledge** $\ell_\infty$ attacks on MNIST. The higher the better. The second from right most column visualises crafted adversarial examples on an image of digit "7", with $\ell_\infty$ distortion $\epsilon$ growing from 0.1 to 0.5.

The adversarial attacks in test include both $\ell_\infty$ and $\ell_2$ untargeted attacks from CleverHans 2.0 library (Papernot et al., 2017a): fast gradient sign method (FGSM, Goodfellow et al., 2014), projected gradient descent (PGD, Madry et al., 2018), momentum iterative attack (MIM, Dong et al., 2017) and Carlini & Wagner $\ell_2$ (CW, Carlini & Wagner, 2017a). Two metrics are reported: *accuracy* of the classifier on crafted adversarial examples, and *detection rate* on adversarial examples that have successfully caused the classifier to misclassify. This detection rate is defined as the true positive (TP) rate of finding an adversarial example, and the detection threshold is selected to achieve a 5% false positive rate on clean training data.

The experiments are performed under various threat model settings. We further evaluate the transferability of crafted adversarial examples across different classifiers in the same way as done in Papernot et al. (2016a). We only provide visualisations in the main text; full table results can be found in the appendix. Readers are also referred to the appendix for further experiments, including a quantitative analysis of the bottleneck effect on robustness and detection.

## 4.1 MNIST

The first set of experiments evaluate the robustness of generative classifiers on MNIST. Here the image pixel values are normalised to $[0, 1]$, and the LVM-based classifiers have $\dim(\boldsymbol{z}) = 64$. We first perform white-box *zero-knowledge* attacks, i.e. the attacker can differentiate through the classifier to craft adversarial examples, but he/she is not aware of the existence of the detector. Then we perform white-box *perfect-knowledge* attacks, where the attacker can differentiate through both the classifier and the detector, and he/she knows the usage of random $\boldsymbol{z}$ samples by the VAE-based classifiers (Biggio et al., 2013; Carlini & Wagner, 2017b). Lastly we consider grey-box and black-box attacks, and evaluate the robustness of generative classifiers against transferred attacks.

**White-box attacks (zero-knowledge, $\ell_\infty$)**   Results for $\ell_\infty$ attacks are reported in Figure 3, and in general generative classifiers perform better in terms of victim accuracy and minimum perturbation of the attacks.[1] By contrast, DFX & DFZ are not robust to the weakest attack (FGSM) even when $\epsilon = 0.2$ (where the adversarial examples are still visually close to the original digit "7"). Interestingly, DBX is the most robust against FGSM & MIM[2], which agrees with the preliminary tests in Alemi et al. (2017). But DBX is less robust to PGD, and here GBZ is a clear winner. These results show that the bottleneck is sometimes beneficial for better robustness of MNIST classifiers.

For detection, generative classifiers have successfully detected the adversarial examples with $\epsilon \geq 0.3$, which is reasonable as the visual distortion is already significant. Importantly, generative classifier's victim accuracy decreases as the $\ell_\infty$ distortion $\epsilon$ increases, but at the same time the TP rates

---

[1]The minimum perturbation is computed across all $\epsilon$ settings. If none of the attack is successful on an input, we manually assign the minimum perturbation of that input as $\epsilon_{\max} + 0.1$ (0.6 for MNIST).

[2]Note that MIM on DBX with $\epsilon = 0.9$ achieves 100% success rate (i.e. victim accuracy 0%).

Table 1: **White-box zero-knowledge** CW-$\ell_2$ attack results. Here, accuracy (adv) measures classifying the adversarial inputs to the original classes.

| | acc. (clean) | acc. (adv) | $\ell_2$ dist. | TP KL |
|---|---|---|---|---|
| BNN | 99.12% | 24.40% | 2.129 | 95.31 |
| GFZ | 98.55% | 28.58% | 2.663 | 95.37 |
| GBZ | 97.45% | 81.51% | 2.446 | 91.01 |
| GFY | 99.15% | 28.64% | 2.732 | 96.03 |
| GBY | 98.72% | 32.72% | 2.735 | 94.46 |
| DFX | 99.10% | 20.31% | 2.095 | 99.96 |
| DBX | 98.87% | 30.19% | 1.806 | 96.76 |
| DFZ | 99.10% | 13.60% | 2.188 | 99.57 |

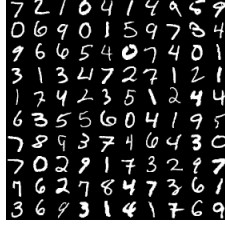 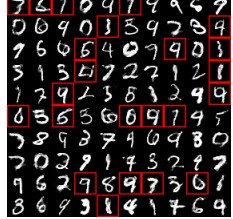

(a) clean inputs     (b) CW adv. inputs

Figure 4: Visualising the clean inputs and the CW adversarial examples crafted on GFZ, digits in red rectangles show significant ambiguity.

for marginal and logit detection also increase. Therefore these $\ell_\infty$ attacks fail to find near-manifold adversarial examples that fool both the classifier and the detection methods. DFZ, as an intermediate between generative and discriminative classifiers, has worse robustness results, but has good detection performance for the marginal and logit metrics. This is because with softmax activation, the marginal distribution $p(\boldsymbol{x})$ is dropped, but in marginal/logit detection $p(\boldsymbol{x})$ is still in use.

**White-box attack (zero-knowledge, $\ell_2$)** For the CW $\ell_2$ attack, we performed a hyper-parameter search for the loss-balancing parameter $c$ ($\ell_2$ distortion increases with larger $c$) in $\{0.1, 1, 10, 100, 1000\}$ on the first 100 test images, and we found $c = 10$ returns the highest success rate. Results are reported in Table 1. Although being successful on fooling many classifiers, CW failed on attacking GBZ. Also, the mean $\ell_2$ distortions of the successful attacks on generative classifiers are significantly larger. Furthermore, we found the success of the attack on other generative classifiers is mainly due to the ambiguity of the crafted adversarial images. As visualised in Figure 4 (also see Figure D.1 in appendix), the induced distortion from CW leads to ambiguous digits which sit at the perceptual boundary between the original and the adversarial classes. With the KL detection method, all classifiers achieve $> 95\%$ detection rates, which is as expected as the default CW attack configuration, by construction, generates adversarial examples that lead to minimal difference between the logit values of the most and the second most probable classes.

**White-box attack (perfect-knowledge, $\ell_\infty$)** The PGD-based perfect-knowledge $\ell_\infty$ attack is designed following (Carlini & Wagner, 2017b): we construct an (approximate) Bayes classifier $p_k(\boldsymbol{y}|\boldsymbol{x})$ using (3) for each set of samples $\{\boldsymbol{z}_c^k\}_{c=1}^C$, and minimize the following with PGD:

$$\mathcal{L}(\boldsymbol{\eta}) = \sum_{k=1}^{K} \log p_k(\boldsymbol{y}|\boldsymbol{x} + \boldsymbol{\eta}) + \lambda_{\text{detect}} \max(0, \Phi(\boldsymbol{x} + \boldsymbol{\eta}, \boldsymbol{y}) - \delta). \tag{4}$$

The detection statistic $\Phi(\boldsymbol{x} + \boldsymbol{\eta}, \boldsymbol{y})$ is $- \log p(\boldsymbol{x} + \boldsymbol{\eta})$ for marginal detection, and $\delta$ is the corresponding threshold computed on training data. For logit/KL detection, the detection statistics and thresholds are constructed accordingly. We label the three attacks against the marginal, logit and KL based detection schemes '-PKM', '-PKL' and '-PKK' respectively. When we only have knowledge of the $K$ samples and not the detection method we call this attack '-PK0' (ie $\lambda_{\text{detect}}$=0 in Eq. 4).

Results are visualised in Figure 5. We see that although the attacker can reduce detection levels, this comes with the trade-off of increasing accuracy, suggesting that a perfect-knowledge adversary cannot break both the classifier and detector working in tandem.

**Sanity checks on gradient masking** Athalye et al. (2018) claimed that if a successful defence against white-box attacks is due to gradient masking, then this defence is likely to be less effective against grey-box/black-box attacks, as they do not differentiate through the victim classifier and the defence mechanism (Papernot et al., 2017b). Therefore, we consider two transfer attacks based on distilling the victim classifier using a "student" CNN which has no gradient masking. The two attacks differ in their threat models: in the **grey-box** setting the attacker has access to the output probability vectors of the classifiers on the training data, while in the **black-box** setting the attacker has access to queried labels only. For the latter black-box setting, we follow Papernot et al. (2017b) to train a substitute CNN using Jacobian-based dataset augmentation, and we refer to appendix B.1 for the detailed algorithm. The grey-box substitutes achieve $> 99\%$ agreement with the victims on test data, and the black-box substitutes obtained $\sim 96\%$ accuracy on test data.

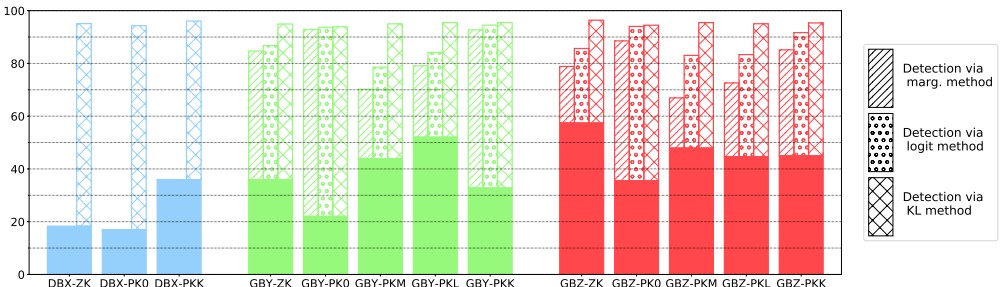

Figure 5: Accuracy and detection rates of DBX, GBY, and GBZ against PGD-based **perfect-knowledge** attack ($\epsilon = 0.2$) on MNIST. The solid area denotes accuracy and the hatched area denotes detection rate with each considered detector. Zero knowledge attacks are labelled '-ZK', other attack labels are described in the main text.

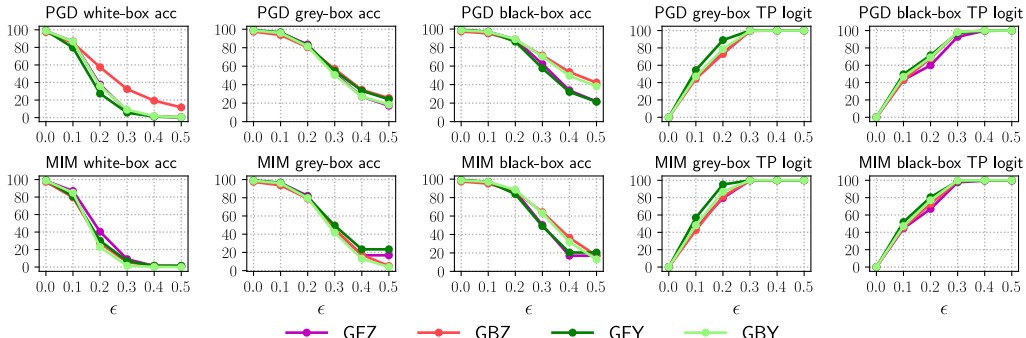

Figure 6: Accuracy and detection rates against **distillation-based** attacks on MNIST. The higher the better. We only present generative classifiers' results here, for full results see appendix E.

Figure 6 shows the accuracy and detection metrics on transferred $\ell_\infty$ attacks crafted on the substitute models, with a comparison to their white-box counterparts. Note that these crafted attacks achieve $\sim$ 100% success rates on fooling the *substitute models* when $\epsilon \geq 0.2$ (see appendix E). However, they do not transfer very well to the generative classifiers. Importantly, for a fixed $\epsilon$ setting, the white-box attacks achieve significantly higher success rates (i.e. lower victim accuracies) than their grey-/black-box counterparts, and the gap is at least $> 20\%$ for $\epsilon \leq 0.3$ (see Table 2). We further present the mean minimum $\ell_\infty$ perturbation in Table 2, and we see that the minimum perturbation obtained by grey-/black-box attacks are significantly higher than those obtained by white-box attacks.

All these results suggest that the robustness of generative classifiers is unlikely to be caused by gradient masking. Again on the detection side, the detection rates increase as the perturbation size increase, and they are near 100% for $\epsilon = 0.3$. This means that most of the successfully transferred adversarial images are off the generative classifier's manifold (as a proxy to the data manifold).

**SPSA (evolutionary strategies)** We consider another black-box setting that only assumes access to the logit values of the prediction given an input. We use the SPSA $\ell_\infty$ attack (Uesato et al., 2018), which is similar to the white-box zero-knowledge attacks, except that gradients are numerically

Table 2: Mean minimum $\ell_\infty$ perturbation (in red, computed on $\epsilon \in \{0.1, 0.2, 0.3, 0.4, 0.5\}$) and victim accuracy (in blue, for $\epsilon \leq 0.3$) for $\ell_\infty$ attacks on MNIST. We manually assign the min. perturbation $\epsilon = 0.6$ to inputs that all attacks failed to find adversarial perturbations.

| Attack | GFZ | GBZ | GFY | GBY |
|---|---|---|---|---|
| PGD(white-box) | 0.23 / 7.71% | 0.30 / 30.78% | 0.21 / 5.52% | 0.23 / 8.89% |
| MIM(white-box) | 0.24 / 9.02% | 0.21 / 4.97% | 0.22 / 6.72% | 0.21 / 1.54% |
| PGD(grey-box) | 0.37 / 51.08% | 0.36 / 50.64% | 0.38 / 53.29% | 0.36 / 48.66% |
| MIM(grey-box) | 0.34 / 43.00% | 0.33 / 40.94% | 0.34 / 46.64% | 0.33 / 40.06% |
| PGD(black-box) | 0.40 / 61.93% | 0.42 / 66.75% | 0.38 / 56.35% | 0.43 / 68.50% |
| MIM(black-box) | 0.36 / 50.44% | 0.38 / 59.86% | 0.36 / 48.07% | 0.39 / 61.78% |

Table 3: Accuracy and detection rates against **black-box SPSA attack** ($\epsilon = 0.3$), with a comparison to white-box PGD on 1000 randomly sampled test datapoints. The higher the better.

| | GFZ | GBZ | GBY | GFY | DFX | DBX | DFZ |
|---|---|---|---|---|---|---|---|
| PGD victim acc | 4.0% | 29.7% | 7.4% | 2.3% | 0.0% | 5.4% | 0.0% |
| SPSA victim acc | 68.2% | 79.0% | 71.0% | 55.9% | 0.0% | 46.3% | 11.0% |
| SPSA TP logit | 91.5% | 92.4% | 95.7% | 98.5% | N/A | N/A | 65.8% |

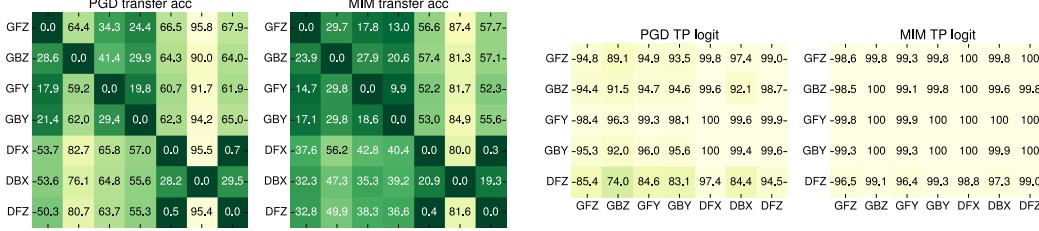

Figure 7: Results on **cross-model transfer** attacks on MNIST. The horizontal axis corresponds to the source victim that the adversarial examples are crafted on, and the vertical axis corresponds to the target victim that the attacks are transferred to. The higher (i.e. the lighter) the better.

estimated using the logit values from the victim classifier. Results in Table 3 clearly show that SPSA performs much worse on generative classifiers when compared to *white-box* PGD. Again this means gradient masking is unlikely to be responsible for the improved robustness of generative classifiers, as utilising the exact gradient yielded improved results.

**Cross-model attack transferability**  Finally, we report in Figure 7 the transferability results of the crafted adversarial examples between different models (Papernot et al., 2016a). Here we take adversarial examples crafted in the white-box setting with PGD and MIM ($\epsilon = 0.3$), and transfer *successful* attacks to other classifiers. The transfer is effective between generative classifiers but not from generative to discriminative (and vice versa). The attacks crafted on DBX do not transfer in general, while at the same time, DBX is the least robust model in this case. Furthermore, the generative classifiers obtain very high detection rates on all transferred attacks ($> 95\%$). In summary, generative classifiers are more robust against the tested transfer attacks across different models.

## 4.2  CIFAR PLANE-VS-FROG BINARY CLASSIFICATION

We consider the same set of evaluations on CIFAR-10, in order to validate the robustness of generative classifiers on natural images (c.f. Carlini & Wagner, 2017b). Unfortunately, we failed to train *fully* generative classifiers with comparable test accuracies to discriminative CNNs (typically $> 80\%$): the clean accuracies for GFZ & GFY are all $< 50\%$. Even when using the conditional PixelCNN++ (Salimans et al., 2017) (which uses much deeper networks), the clean accuracy on the test data is $72.4\%$. Instead, we consider a simpler binary classification problem and construct a dataset containing CIFAR images from the "airplane" and "frog" categories. The images in this dataset are scaled to [0, 1]. On this dataset, the generative classifiers use $\dim(\boldsymbol{z}) = 128$ and obtain $> 90\%$ clean test accuracy (see appendix). The attacks are performed on the test images that all models initially correctly classify, leading to a test set of 1577 instances. Due to the page limit, we only present white-box zero-knowledge attacks here, and discuss further attacks in the appendix.

**White-box attacks (zero-knowledge, $\ell_\infty$)**  We present the white-box $\ell_\infty$ attack results in Figure 8, where the distortion strengths are selected as $\epsilon \in \{0.01, 0.02, 0.05, 0.1, 0.2\}$. Again, generative classifiers are more robust than the discriminative ones, and GBZ is the most robust, much better than the others when $\epsilon \geq 0.05$. BNN is significantly better than other discriminative VAE-based classifiers, presumably due to higher randomness. Detection results are less satisfactory: marginal/logit detection fail to detect attacks with $\epsilon = 0.1$ (which attain both high success rate and induce visually perceptible distortion). KL detection performs better, and interestingly, discriminative classifiers dominate in this metric. These results suggest that the $\ell_2$ likelihood might not be best suited for modelling natural images (c.f. Larsen et al., 2016; van den Oord et al., 2016). Still, the minimum distortion required to fool generative classifiers are much higher than that for discriminative ones, indeed the visual distortion of the adversarial examples on generative classifiers are more significant.

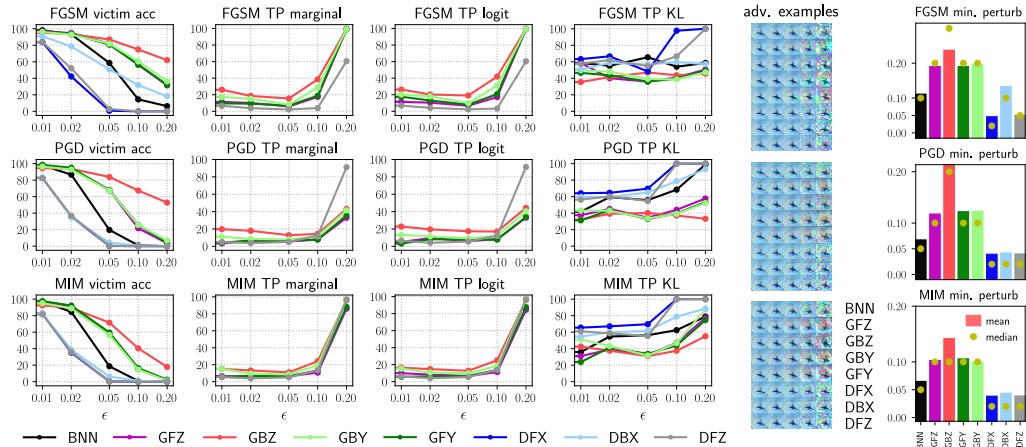

Figure 8: Accuracy, detection rates and minimum $\ell_{\inf}$ perturbations against **white-box zero-knowledge** $\ell_{\infty}$ attacks on the CIFAR plane-vs-frog dataset. The higher the better. The second from right most column visualises crafted adversarial examples on an image of a plane, with $\ell_{\infty}$ distortion $\epsilon \in \{0.01, 0.02, 0.05, 0.1, 0.2\}$.

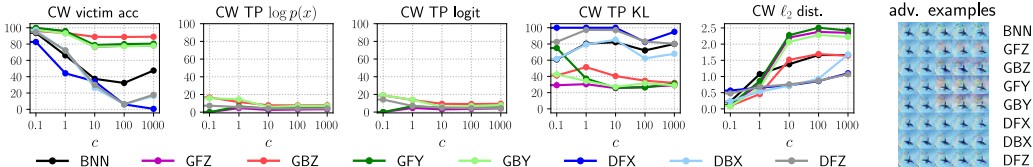

Figure 9: Accuracy and detection rates against **white-box zero-knowledge** CW attacks on the CIFAR plane-vs-frog dataset. The higher the better.

**White-box attack (zero-knowledge, $\ell_2$)** As the test set is relatively small, we directly perform CW attacks on the test data with $c \in \{0.1, 1, 10, 100, 1000\}$. Results are visualised in Figure 9. The generative classifiers are significantly more robust than the others (with the best being GBZ), and the mean $\ell_2$ distortions computed on successful attacks are also significantly higher. The TP rates are low for marginal and logit detection, which is reasonable as the crafted images are visually similar to the clean ones. Note that the distortion for the attacks on generative classifiers is perceptible. These results indicate that this CW attack is ineffective when attacking generative classifiers.

### 4.3 FULL CIFAR-10: COMBINING DEEP BAYES AND DISCRIMINATIVE FEATURES

The final experiment examines the robustness of CIFAR-10 *multi-class* classifiers, with the generative classifiers trained on *discriminative* visual features. To do this, we download a pretrained deep convolutional net[3] with VGG16-like architecture (Simonyan & Zisserman, 2014), and use its feature representation $\phi(\boldsymbol{x})$ as the input to the VAE-based classifiers: $p(\boldsymbol{y}|\boldsymbol{x}) = p(\boldsymbol{y}|\phi(\boldsymbol{x}))$. The classifiers in test include GBZ, GBY and DBX. We use fully-connected neural networks for these classifiers, and select from VGG16 the 9th convolution layer (CONV9) and the first fully connected layer after convolution (FC1) as the feature layers to ensure $\sim 90\%$ test accuracy (see appendix).

Results on **white-box zero-knowledge** $\ell_{\infty}$ **attacks** are visualised in Figure 10. Compared with the VGG16 baseline, we see clear improvements in robustness and detection for all VAE-based classifiers. In particular, the generative classifiers GBZ and GBY are overall better than DBX. More importantly, generative classifiers based on CONV9 features are significantly more robust than those based on FC1 features. In contrast, for DBX, which is *discriminative*, the robustness results are very similar, indicating that the level of feature representation has little effect. These results suggest that one can achieve both high clean accuracy and better robustness against adversaries by combining discriminatively learned visual features and generative classifiers.

---

[3] `https://github.com/geifman/cifar-vgg`, 93.59% test accuracy

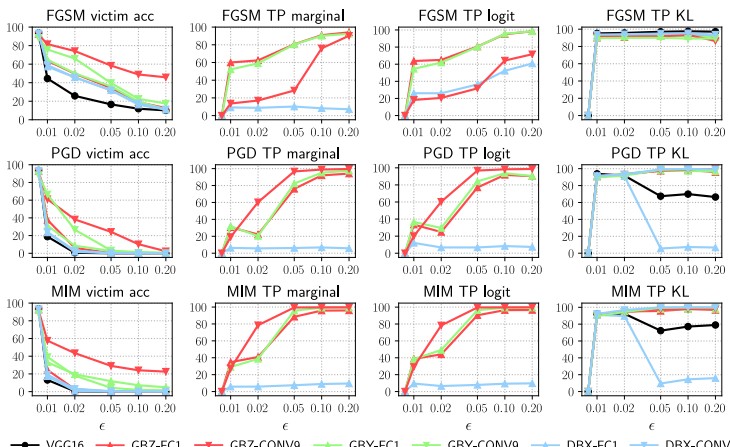

Figure 10: Accuracy and detection rates against **white-box zero-knowledge** $\ell_\infty$ attacks on CIFAR-10. The higher the better. Note that results for the DBX classifiers are almost identical.

## 5 DISCUSSION

We have proposed *deep Bayes* as a generative classifier that uses deep latent variable models to model the joint distribution of input-output pairs. We have given evidence, on multiple classification tasks, that generative classifiers are more robust to adversarial attacks than discriminative classifiers. Furthermore, the logit in generative classifiers has a well defined meaning and can be used to detect attacks, even when the classifier is fooled.

Our results corroborate with the Bayesian neural network literature, in particular (Li & Gal, 2017; Feinman et al., 2017; Carlini & Wagner, 2017b), in showing that modelling unobserved variables are effective for defending against adversarial attacks[4]. Concurrent to us, Schott et al. (2018) also demonstrated the robustness of generative classifiers on MNIST, in which the logits are computed by a tempered version of the variational lower-bound. However, their approach requires thousands of random $z$ samples and tens of optimisation steps to approximate $\log p(x|y)$ for every input-output pair $(x, y)$, making it less scalable than our importance sampling technique to large datasets and big architectures. Indeed, we have scaled our approach to CIFAR-10, a natural image dataset, and the robustness results are consistent with those on MNIST. In addition, we have also shown that the structure of the graphical model has a significant impact on robustness: deep LVM-based generative classifiers generally outperform the (randomised) discriminative ones.

While we have given strong evidence to suggest that generative classifiers are more robust to current adversarial attacks, we do not wish to claim that these models will be robust to *all* possible attacks. Aside from many recent attacks being designed specifically for discriminative neural networks, there is also evidence for the fragility of generative models; e.g. naive Bayes as a standard approach for spam filtering is well-known to be fragile (Dalvi et al., 2004; Huang et al., 2011), and very recently Tabacof et al. (2016); Kos et al. (2017); Creswell et al. (2017) also designed attacks for (unconditional) VAE-type models. However, generative classifiers can be made more robust too, to counter these weaknesses. Dalvi et al. (2004) have shown that generative classifiers can be made more secure if aware of the attack strategy, and Biggio et al. (2011; 2014) further improved naive Bayes' robustness by modelling the conditional distribution of the adversarial inputs. These approaches are similar to the adversarial training of discriminative classifiers, and efficient ways for doing so with generative classifiers can be an interesting research direction.

But even with this note of caution, we believe this work offers exciting avenues for future work. Using generative classifiers offers an interesting way to evaluate generative models and can drive improvements in their ability to tackle high-dimensional datasets, where traditionally generative classifiers have been less accurate than discriminative classifiers (Efron, 1975; Ng & Jordan, 2002). In addition, the combination of generative and discriminative models investigated in this paper is a compelling direction for future research. Overall, we believe that progress on generative classifiers can inspire better designs of attack, defence and detection techniques.

---

[4]In Bayesian neural networks, the network weights are treated as unobserved/latent variables.

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

## A    MODEL ARCHITECTURES

**MNIST experiments**    The VAEs are constructed with convolutional encoders and deconvolutional generators. More specifically, the encoder network for $q(z|x, y)$ is the same across all VAE-based classifiers. It starts with a 3-layer convolutional neural network with $5 \times 5$ filters and 64 channels, with a max-pooling operation after each convolution. Then, the convolutional network is followed by a MLP with 2 hidden layers, each with 500 units, to produce the mean and variance parameters of $q$. The label $y$ is injected into the MLP at the first hidden layer, as a one hot encoding (i.e. for MNIST, the first hidden layer has 500+10 units). The latent dimension is $\dim(z) = 64$.

The $p$ models' architectures are the following:

- GFZ: For $p(y|z)$ we use a MLP with 1 hidden layer composed of 500 units. For $p(x|y, z)$ we used an MLP with 2 hidden layers, each with 500 units, and $4 \times 4 \times 64$ dimension output, followed by a 3-layer deconvolutional network with $5 \times 5$ kernel size, stride 2 and [64, 64, 1] channels.

- GFY: We use an MLP with 1 hidden layer composed of 500 units for $p(z|y)$, and the same architecture as GFZ for $p(x|y, z)$.

- DFZ: We use almost the same deconvolutional network architecture for $p(x|z)$ as GFZ's $p(x|y, z)$ network, except that the input is $z$ only. For $p(y|x, z)$ we use almost the same architecture as $q(z|x, y)$ except that the injected input to the MLP is $z$ and the MLP output is the set of logit values for $y$.

- DFX: We use the same architecture as G3 for $p(y|x, z)$. The network for $p(z|x)$ is almost identical except that there is no injected input to the MLP, and the network returns the mean and variance parameters for $q(z|x)$.

- DBX: We use GFZ's architecture for $p(y|z)$ and DFX's architecture for $p(z|x)$.

- GBY: We use GFY's architecture for $p(z|y)$ and DFZ's architecture for $p(x|z)$.

- GBZ: We use GFZ's architecture for $p(y|z)$ and DFZ's architecture for $p(x|z)$.

The BNN has almost the same architecture as the encoder network $q$, except that it uses 2x the hidden units/channels, and the last layer is 10 dimensions. Note that here we used dropout as it is convenient to implement, and we expect better approximate inference methods (such as stochastic gradient MCMC) to return better results for robustness and detection.

**CIFAR plane-vs-frog experiments**    The model architectures are almost the same as used in MNIST experiments, except that the hidden layer dimensions for the MLP layers are increased to 1000. For the encoder $q$, the channels are increased to [64, 128, 256]. For the $p$ models, the deconvolutional networks have different channel values, [128, 64, 3], and the MLP before the deconvolution outputs a $4 \times 4 \times 256$ vector (before reshaping). The BNN has 2x the channels but still uses 1000 hidden units.

**CIFAR-10 experiments**    The pre-trained VGG16 network is downloaded from `https://github.com/geifmany/cifar-vgg`, where the CONV9 and FC1 layers correspond to:

CONV9: `https://github.com/geifmany/cifar-vgg/blob/master/cifar10vgg.py#L82`

FC1: `https://github.com/geifmany/cifar-vgg/blob/master/cifar10vgg.py#L109`

The VAE-based classifiers build fully connected networks on top of the extracted features, and use $\dim(z) = 128$ for bottleneck. The encoder $q(z|\phi(x), y)$ has the network architectures $[\dim(\phi(x)) + \dim(y), 1000, 1000, \dim(z) \times 2]$, and we use the same encoder architecture across all classifiers. The decoder architectures are as follows:

- DBX: We use an MLP of layers $[\dim(z), 1000, \dim(y)]$ for $p(y|z)$ and an MLP of layers $[\dim(\phi(x)), 1000, 1000, \dim(z) \times 2]$ for $p(z|\phi(x))$.

GBZ: We use an MLP of layers [dim($z$), 1000, 1000, dim($y$)] for $p(y|z)$ and an MLP of layers [dim($z$), 1000, 1000, dim($\phi(x)$)] for $p(\phi(x)|z)$.

GBY: We use an MLP of layers [dim($y$), 1000, dim($z$) $\times$ 2] for $p(z|y)$ and GBZ's architecture for $p(\phi(x)|z)$.

## B  ATTACK SETTINGS

We use the Cleverhans package to perform attacks. We use the default hyper-parameters, if not specifically stated.

**PGD:** We perform the attack for 40 iterations with step-size 0.01.

**MIM:** We perform the attack for 40 iterations with step-size 0.01 and decay factor 1.0.

**CW-$\ell_2$:** We use learning rate 0.01 and confidence 0, and we optimise the loss for 1000 iterations.

**SPSA:** We use almost the same hyper-parameters as in Uesato et al. (2018) except for the number of samples for gradient estimates. In detail, we perform the attack for 100 iterations with perturbation size 0.01, Adam learning rate 0.01, stopping threshold -5.0 and 2000 samples for each gradient estimate.

### B.1  JACOBIAN-BASED DATASET AUGMENTATION

The black-box distillation attack is based on Papernot et al. (2017b), which trains a substitute CNN using Jacobian-based dataset augmentation. Assume $y = F(x)$ is the output one-hot vector of the victim, and $p(x)$ is the probability vector output of the substitute model, then at the $t^{\text{th}}$ outer-loop, we train the substitute CNN on dataset $\mathcal{D}_t = \{(x_n, y_n)\}$ with queried $y_n$ for 10 epochs, and augment the dataset by

$$\mathcal{D}_{t+1} = \mathcal{D}_t \cup \{(\hat{x}, F(\hat{x})) \mid \hat{x} = x + \lambda \nabla_x p(x)^{\text{T}} y, \ (x, y) \in \mathcal{D}_t\}. \tag{5}$$

We initialise $\mathcal{D}_1$ with $200 \times 10$ datapoints from the MNIST test set, select $\lambda = 0.1$, and run the algorithm for 6 outer-loops. On MNIST, this results in $64,000$ queried inputs, and $\sim 96\%$ accuracy of the substitute model on test data. On CIFAR binary classification, we use $200 \times 2$ datapoints for the inital query set $\mathcal{D}_1$, resulting in $12,800$ queries in total. The substitutes achieved almost the same accuracy as their corresponding victim models on clean test datapoints.

## C  FURTHER EXPERIMENTS

### C.1  FURTHER EXPERIMENTS ON CIFAR BINARY CLASSIFICATION

**White-box attack (perfect-knowledge, $\ell_\infty$)**  Figure C.1 shows the perfect knowledge attack on the CIFAR binary classification task. Again we see that although the attack is effective for the detection schemes, it comes with the price of decreased mis-classification rates. Interestingly GBY seems to be robust to this attack, where the accuracies on the crafted adversarial examples increase.

**Sanity checks on gradient masking**  We conducted the same sets of transferred $\ell_\infty$ attack experiments, and presents the results in Figure C.2 and Table C.1. Again that these crafted attacks achieve $\sim 100\%$ success rates on fooling the *substitute models* when $\epsilon \geq 0.1$. Similar to the MNIST experiments, these adversarial examples do not transfer very well to the generative classifiers, and for a fixed $\epsilon$ setting, the white-box attacks achieve significantly higher success rates than their grey-/black-box counterparts (with the gap at $\epsilon \leq 0.1$ around 30%). Furthermore, the minimum perturbation obtained by grey-/black-box attacks are significantly higher than those obtained by white-box attacks. For detection, the detection rates are relatively low at $\epsilon \leq 0.1$ (where the classifiers achieved high accuracy). But the detection rates increase significantly for $\epsilon = 0.2$, where the victim accuracies also drop.

All these results suggest that the robustness of generative classifiers is unlikely to be caused by gradient masking. Also most of the successfully transferred adversarial images are off the generative classifier's manifold (as a proxy to the data manifold).

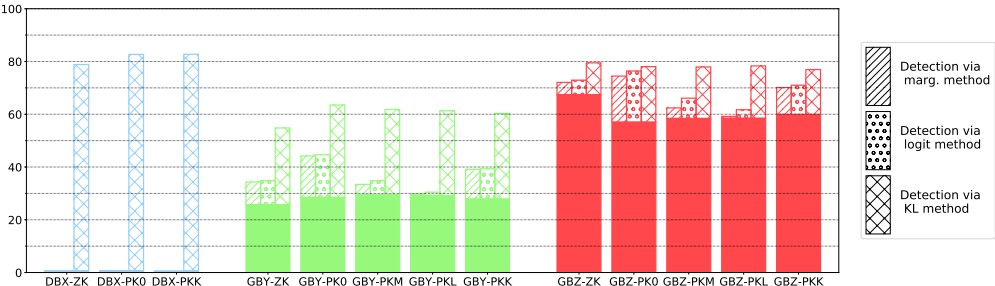

Figure C.1: Accuracy and detection rates of DBX, GBY, and GBZ against PGD-based **perfect-knowledge** attack ($\epsilon = 0.1$) on CIFAR binary task. The solid area denotes accuracy and the hatched area denotes detection rate with each considered detector. Zero knowledge attacks are labelled '-ZK', other attack labels are the same as for the MNIST plot (Figure 5).

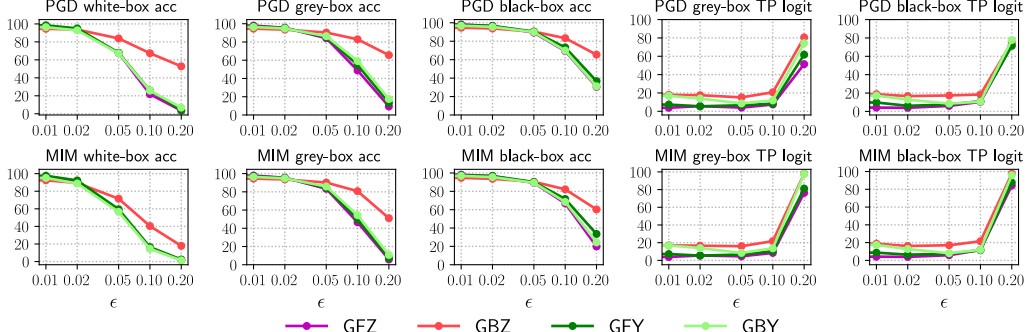

Figure C.2: Accuracy and detection rates against **distillation-based** attacks on CIFAR plane-vs-frog binary classification. The higher the better. We only present generative classifiers' results here, for full results see appendix E.

Table C.1: Mean minimum $\ell_\infty$ perturbation (in red, computed on $\epsilon \in \{0.01, 0.02, 0.05, 0.1, 0.2\}$) and victim accuracy (in blue, for $\epsilon \leq 0.1$) for $\ell_\infty$ attacks on CIFAR plane-vs-frog binary classification. We manually assign the min. perturbation $\epsilon = 0.3$ to inputs that all attacks failed to find adversarial perturbations.

| Attack | GFZ | GBZ | GFY | GBY |
|---|---|---|---|---|
| PGD(white-box) | 0.11 / 21.81% | 0.20 / 65.63% | 0.11 / 25.81% | 0.11 / 25.24% |
| MIM(white-box) | 0.09 / 15.22% | 0.13 / 37.60% | 0.10 / 16.4%9 | 0.09 / 14.39% |
| PGD(grey-box) | 0.15 / 50.48% | 0.23 / 77.30% | 0.16 / 54.66% | 0.17 / 57.96% |
| MIM(grey-box) | 0.15 / 47.62% | 0.21 / 75.71% | 0.15 / 51.11% | 0.16 / 53.84% |
| PGD(black-box) | 0.19 / 68.36% | 0.23 / 79.45% | 0.20 / 70.13% | 0.19 / 67.98% |
| MIM(black-box) | 0.18 / 66.39% | 0.23 / 78.38% | 0.19 / 68.42% | 0.19 / 66.52% |

Table C.2: Accuracy and detection rates against **black-box SPSA attack** ($\epsilon = 0.05$) on CIFAR plane-vs-frog, with a comparision to white-box PGD. The higher the better.

|  | GFZ | GBZ | GBY | GFY | DFX | DBX | DFZ |
|---|---|---|---|---|---|---|---|
| PGD victim acc | 67.7% | 83.9% | 67.9% | 67.3% | 0.3% | 4.2% | 0.4% |
| SPSA victim acc | 96.4% | 95.2% | 96.4% | 96.3% | 0.4% | 87.5% | 5.1% |
| SPSA TP logit | 10.0% | 17.1% | 15.5% | 10.0% | N/A | N/A | 4.5% |

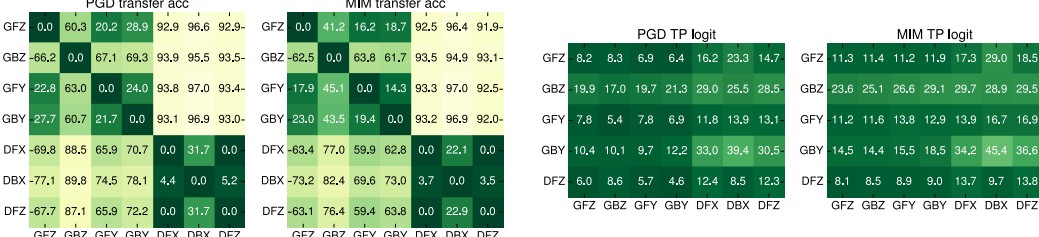

Figure C.3: Results on **cross-model transfer** attacks on the CIFAR plane-vs-frog dataset. The horizontal axis corresponds to the source victim, and the vertical axis corresponds to the target victim. The higher (lighter) the better.

**SPSA (evolutionary strategies)**  Similarly we perform the SPSA attack (Uesato et al., 2018) on the CIFAR binary classification task. The results for $\epsilon = 0.05$ are presented in Table C.2, with a comparison to the *white-box* PGD attacks. Again we see that SPSA fails to attack generative classifiers, and the bottleneck discriminative classifier DBX is significantly more robust than the discriminative ones with fully-connected graphical models.

**Cross-model attack transferability**  Finally, we present the CIFAR cross-model attack transferability results in Figure C.3, and here we select $\epsilon = 0.1$ instead. Again, transferred attacks are less effective across the victim models. However, the TP rates for logit detection are significantly lower than in the MNIST case (also see Figure 8). Nevertheless, the detection rates for the "discriminative to generative" transfer are considerably higher. Combined with the accuracy results, we see that discriminative models as substitutes are ineffective in the transferred attack setting.

## C.2 QUANTIFYING THE EFFECT OF THE BOTTLENECK LAYER

We see from the main text that classifiers with bottleneck structure may be preferred for resisting adversarial examples. To quantify this bottleneck effect, we train on MNIST models DBX, GBZ and GBY with $z$ dimensions in $\{16, 32, 64, 128\}$ (the main text experiments use $\dim(z) = 64$).The clean test accuracy is shown in Table C.3, showing that all models in test perform reasonably well.

Table C.3: Clean test accuracy on MNIST classification (with varied bottleneck layer sizes).

|  | $\dim(z) = 16$ | $\dim(z) = 32$ | $\dim(z) = 64$ | $\dim(z) = 128$ |
|---|---|---|---|---|
| DBX | 99.11% | 99.01% | 98.98% | 98.91% |
| GBZ | 97.11% | 97.08% | 97.45% | 96.62% |
| GBY | 98.82% | 98.95% | 98.72% | 98.75% |

We repeat the same **white-box zero-knowledge** $\ell_\infty$ attack experiments as done in the main text, where results are presented in Figure C.4 and Tables E.25, E.26 and E.27. It is clear that for discriminative classifiers, DBX, the models become less robust as the bottleneck dimension $\dim(z)$ increases. Interestingly DBX classifiers seem to be very robust against FGSM attacks, which agrees with the results in Alemi et al. (2017). For the generative ones, we also observe similar trends (although much less significant) of decreased robustness for GBY classifiers, and for GBZ the trend is unclear, presumably due to local optimum issues in optimisation. In summary, GBZ classifiers are generally more robust compared to GBY classifiers. More importantly, when the accuracy of

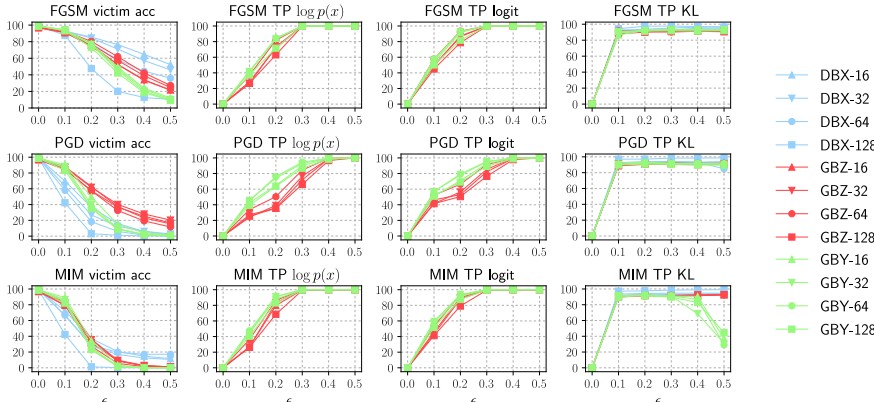

Figure C.4: Accuracy and detection rates against **white-box zero-knowledge** $\ell_\infty$ attacks on MNIST, with varied bottleneck layer sizes.

Table D.1: Clean test accuracy on CIFAR plane-vs-frog classification.

| BNN | GFZ | GFY | DFZ | DFX | DBX | GBZ | GBY |
|---|---|---|---|---|---|---|---|
| 97.00% | 91.60% | 91.20% | 94.85% | 95.65% | 96.00% | 89.35% | 90.65 |

generative classifiers on adversarial images decreases to zero, the detection rates with marginal/logit detection increases to $100\%$. This clearly shows that the three attacks tested here cannot fool the generative classifiers without being detected.

# D ADDITIONAL RESULTS

We visualise in Figure D.1 the crafted adversarial images using white-box CW attack.

We present in Table D.1 the clean accuracy on CIFAR plane-vs-frog test images (2000 in total).

We present in Table D.2 the clean accuracy on CIFAR-10 test images.

Table D.2: Clean test accuracy on CIFAR-10 classification.

| VGG16 | GBZ-FC1 | GBY-FC1 | DBX-FC1 | GBZ-CONV9 | GBY-CONV9 | DBX-CONV9 |
|---|---|---|---|---|---|---|
| 93.59% | 92.55% | 93.21% | 93.49% | 91.76% | 88.33% | 93.21% |

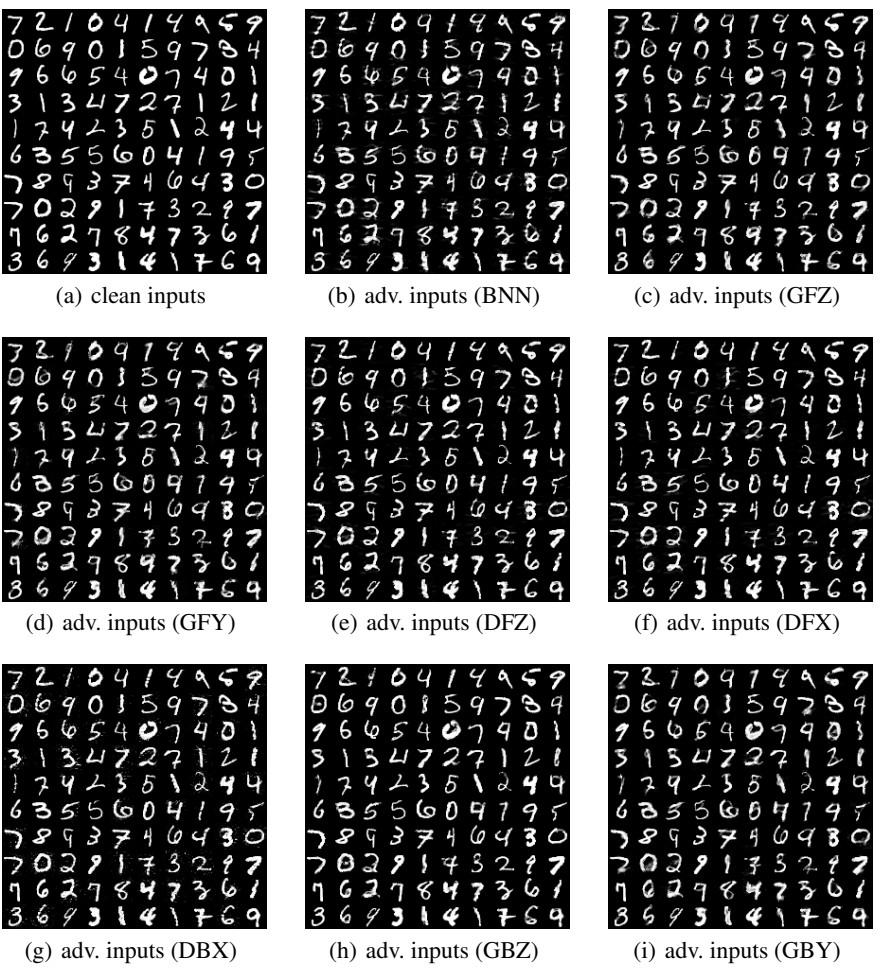

(a) clean inputs      (b) adv. inputs (BNN)      (c) adv. inputs (GFZ)

(d) adv. inputs (GFY)      (e) adv. inputs (DFZ)      (f) adv. inputs (DFX)

(g) adv. inputs (DBX)      (h) adv. inputs (GBZ)      (i) adv. inputs (GBY)

Figure D.1: Visualising the clean inputs of MNIST and the CW adversarial examples crafted on all the classifiers.

# E   RESULTS IN TABLES

We present in tables the full results of the experiments.

See Tables E.1 to E.9 for the white-box attacks.

See Tables E.10 to E.15 for the grey-box attacks.

See Tables E.16 to E.21 for the black-box attacks.

See Tables E.22 to E.24 for CIFAR-10 results with VGG-based classifiers.

See Tables E.25 to E.27 for bottleneck effect quantification results.

Table E.1: FGSM white-box zero-knowledge attack results on MNIST.

| $\epsilon$ | acc. (adv) | | | | | TP marginal | | | | | TP logit | | | | | TP KL | | | | |
|---|---|---|---|---|---|---|---|---|---|---|---|---|---|---|---|---|---|---|---|---|
| | 0.10 | 0.20 | 0.30 | 0.40 | 0.50 | 0.10 | 0.20 | 0.30 | 0.40 | 0.50 | 0.10 | 0.20 | 0.30 | 0.40 | 0.50 | 0.10 | 0.20 | 0.30 | 0.40 | 0.50 |
| BNN | 92.4 | 67.8 | 40.5 | 26.2 | 20.4 | N/A | N/A | N/A | N/A | N/A | N/A | N/A | N/A | N/A | N/A | 93.0 | 94.2 | 95.4 | 95.5 | 96.5 |
| GFZ | 94.2 | 74.5 | 38.9 | 12.9 | 5.7 | 43.6 | 79.8 | 100.0 | 100.0 | 100.0 | 56.4 | 89.6 | 99.9 | 100.0 | 100.0 | 89.2 | 91.7 | 92.2 | 92.5 | 92.2 |
| GBZ | 92.5 | 80.3 | 62.0 | 42.4 | 27.2 | 37.0 | 81.7 | 100.0 | 100.0 | 100.0 | 57.6 | 93.5 | 100.0 | 100.0 | 100.0 | 91.6 | 90.9 | 90.3 | 91.1 | 91.5 |
| GFY | 94.3 | 74.8 | 46.5 | 21.7 | 10.5 | 53.1 | 92.6 | 100.0 | 100.0 | 100.0 | 66.2 | 97.9 | 100.0 | 100.0 | 100.0 | 90.8 | 93.1 | 93.8 | 94.2 | 94.2 |
| GBY | 93.6 | 76.4 | 47.5 | 22.3 | 10.7 | 41.9 | 84.5 | 100.0 | 100.0 | 100.0 | 57.7 | 92.5 | 100.0 | 100.0 | 100.0 | 89.3 | 92.7 | 92.8 | 92.9 | 92.9 |
| DFX | 70.1 | 14.8 | 1.0 | 0.4 | 0.6 | N/A | N/A | N/A | N/A | N/A | N/A | N/A | N/A | N/A | N/A | 93.6 | 93.8 | 93.8 | 93.5 | 94.3 |
| DBX | 91.6 | 77.8 | 58.1 | 44.5 | 36.0 | N/A | N/A | N/A | N/A | N/A | N/A | N/A | N/A | N/A | N/A | 92.6 | 93.6 | 95.2 | 96.3 | 97.0 |
| DFZ | 75.2 | 19.0 | 2.4 | 1.1 | 1.0 | 12.6 | 50.6 | 100.0 | 100.0 | 100.0 | 26.9 | 65.7 | 100.0 | 100.0 | 100.0 | 93.1 | 95.1 | 94.5 | 94.9 | 94.8 |

Table E.2: PGD white-box zero-knowledge attack results on MNIST.

| $\epsilon$ | acc. (adv) | | | | | TP marginal | | | | | TP logit | | | | | TP KL | | | | |
|---|---|---|---|---|---|---|---|---|---|---|---|---|---|---|---|---|---|---|---|---|
| | 0.10 | 0.20 | 0.30 | 0.40 | 0.50 | 0.10 | 0.20 | 0.30 | 0.40 | 0.50 | 0.10 | 0.20 | 0.30 | 0.40 | 0.50 | 0.10 | 0.20 | 0.30 | 0.40 | 0.50 |
| BNN | 83.2 | 12.0 | 0.5 | 0.0 | 0.0 | N/A | N/A | N/A | N/A | N/A | N/A | N/A | N/A | N/A | N/A | 92.1 | 92.1 | 88.7 | 20.4 | 0.2 |
| GFZ | 86.7 | 37.7 | 7.7 | 1.2 | 0.3 | 43.2 | 71.8 | 91.4 | 99.4 | 100.0 | 55.8 | 77.1 | 94.8 | 99.6 | 100.0 | 90.3 | 92.1 | 90.9 | 90.4 | 89.7 |
| GBZ | 85.1 | 57.4 | 32.5 | 19.3 | 11.6 | 33.7 | 50.3 | 84.4 | 99.7 | 100.0 | 52.2 | 66.3 | 91.5 | 99.8 | 100.0 | 90.0 | 91.5 | 91.5 | 91.7 | 91.8 |
| GFY | 79.7 | 27.4 | 5.6 | 1.2 | 0.3 | 58.1 | 87.9 | 98.0 | 100.0 | 100.0 | 68.7 | 92.2 | 99.3 | 100.0 | 100.0 | 92.6 | 92.5 | 90.7 | 90.7 | 85.4 |
| GBY | 86.7 | 35.9 | 9.0 | 1.7 | 0.4 | 45.3 | 76.1 | 94.6 | 99.7 | 100.0 | 56.9 | 79.3 | 95.6 | 99.9 | 100.0 | 90.1 | 92.1 | 91.6 | 91.7 | 91.4 |
| DFX | 47.6 | 0.7 | 0.0 | 0.0 | 0.0 | N/A | N/A | N/A | N/A | N/A | N/A | N/A | N/A | N/A | N/A | 92.8 | 89.8 | 13.3 | 31.4 | 42.8 |
| DBX | 58.0 | 18.3 | 6.0 | 1.3 | 0.2 | N/A | N/A | N/A | N/A | N/A | N/A | N/A | N/A | N/A | N/A | 92.7 | 94.0 | 94.1 | 93.7 | 84.8 |
| DFZ | 49.6 | 1.0 | 0.0 | 0.0 | 0.0 | 11.8 | 44.3 | 94.1 | 100.0 | 100.0 | 25.9 | 57.6 | 94.5 | 99.9 | 100.0 | 93.9 | 89.8 | 12.1 | 22.9 | 28.0 |

Table E.3: MIM white-box zero-knowledge attack results on MNIST.

| $\epsilon$ | acc. (adv) | | | | | TP marginal | | | | | TP logit | | | | | TP KL | | | | |
|---|---|---|---|---|---|---|---|---|---|---|---|---|---|---|---|---|---|---|---|---|
| | 0.10 | 0.20 | 0.30 | 0.40 | 0.50 | 0.10 | 0.20 | 0.30 | 0.40 | 0.50 | 0.10 | 0.20 | 0.30 | 0.40 | 0.50 | 0.10 | 0.20 | 0.30 | 0.40 | 0.50 |
| BNN | 82.0 | 7.2 | 0.1 | 0.0 | 0.0 | N/A | N/A | N/A | N/A | N/A | N/A | N/A | N/A | N/A | N/A | 92.2 | 91.1 | 72.0 | 64.5 | 7.1 |
| GFZ | 87.0 | 40.4 | 9.0 | 1.2 | 1.2 | 43.5 | 83.8 | 98.4 | 99.4 | 99.4 | 57.6 | 89.3 | 98.6 | 99.2 | 99.2 | 91.5 | 92.3 | 92.0 | 90.6 | 90.6 |
| GBZ | 79.6 | 27.4 | 5.6 | 1.5 | 0.5 | 36.1 | 86.3 | 100.0 | 100.0 | 100.0 | 53.5 | 93.1 | 99.9 | 100.0 | 100.0 | 91.3 | 91.4 | 91.9 | 92.2 | 92.7 |
| GFY | 80.8 | 30.1 | 6.8 | 1.4 | 1.4 | 56.0 | 93.7 | 99.8 | 100.0 | 100.0 | 68.1 | 96.6 | 99.9 | 100.0 | 100.0 | 90.9 | 92.4 | 92.0 | 91.6 | 91.6 |
| GBY | 84.9 | 22.9 | 1.5 | 0.1 | 0.0 | 47.4 | 91.2 | 99.9 | 100.0 | 100.0 | 59.8 | 92.8 | 99.9 | 100.0 | 100.0 | 91.4 | 92.1 | 91.2 | 86.8 | 29.0 |
| DFX | 48.4 | 0.8 | 0.0 | 0.0 | 0.0 | N/A | N/A | N/A | N/A | N/A | N/A | N/A | N/A | N/A | N/A | 93.2 | 89.6 | 9.2 | 17.5 | 17.5 |
| DBX | 66.7 | 28.7 | 19.7 | 17.2 | 17.2 | N/A | N/A | N/A | N/A | N/A | N/A | N/A | N/A | N/A | N/A | 93.5 | 94.3 | 94.6 | 94.7 | 94.7 |
| DFZ | 50.5 | 1.2 | 0.0 | 0.0 | 0.0 | 11.9 | 50.7 | 99.4 | 100.0 | 100.0 | 26.4 | 63.2 | 99.0 | 100.0 | 100.0 | 94.3 | 89.8 | 11.5 | 17.3 | 17.3 |

Table E.4: Perfect knowledge attacks on MNIST. This is done using the PGD attack with $\epsilon = 0.2$. 'ZK' stands for zero knowledge, 'PK0' where you have knowledge of the $K$ samples but not of the detection system. 'PKM', 'PKL', and 'PKK' are attacks where you have knowledge of the K samples and of the marginal, logit and KL detection mechanisms respectively.

| | ZK | | | | PK0 | | | | PKM | | | | PKL | | | | PKK | | | |
|---|---|---|---|---|---|---|---|---|---|---|---|---|---|---|---|---|---|---|---|---|
| | acc. | TP marg. | TP logit | TP KL | acc. | TP marg. | TP logit | TP KL | acc. | TP marg. | TP logit | TP KL | acc. | TP marg. | TP logit | TP KL | acc. | TP marg. | TP logit | TP KL |
| DBX | 18.3 | N/A | N/A | 94 | 17 | N/A | N/A | 93.1 | N/A | N/A | N/A | N/A | N/A | N/A | N/A | N/A | 35.9 | N/A | N/A | 93.9 |
| GBZ | 57.4 | 50.3 | 66.3 | 91.5 | 35.5 | 82.2 | 90.7 | 91.4 | 47.9 | 36.5 | 67.4 | 91.3 | 44.6 | 50.5 | 69.9 | 91.0 | 45 | 73.0 | 84.8 | 91.5 |
| GBY | 35.9 | 76.1 | 79.3 | 92.1 | 22 | 90.8 | 91.9 | 92.1 | 43.8 | 47.0 | 61.9 | 91.1 | 52.0 | 56.4 | 66.9 | 90.6 | 32.8 | 89.2 | 91.8 | 93.3 |

Table E.5: FGSM white-box zero-knowledge attack results on CIFAR plane-vs-frog binary classification.

| | acc. (adv) | | | | | TP marginal | | | | | TP logit | | | | | TP KL | | | | |
|---|---|---|---|---|---|---|---|---|---|---|---|---|---|---|---|---|---|---|---|---|
| $\epsilon$ | 0.01 | 0.02 | 0.05 | 0.10 | 0.20 | 0.01 | 0.02 | 0.05 | 0.10 | 0.20 | 0.01 | 0.02 | 0.05 | 0.10 | 0.20 | 0.01 | 0.02 | 0.05 | 0.10 | 0.20 |
| BNN | 98.2 | 93.2 | 58.5 | 14.5 | 6.3 | N/A | N/A | N/A | N/A | N/A | N/A | N/A | N/A | N/A | N/A | 58.6 | 55.1 | 65.5 | 54.0 | 58.5 |
| GFZ | 97.1 | 94.7 | 81.8 | 56.4 | 31.4 | 11.4 | 10.1 | 5.9 | 17.4 | 99.3 | 11.4 | 10.4 | 6.8 | 17.2 | 99.3 | 58.6 | 40.1 | 35.9 | 41.0 | 50.6 |
| GBZ | 95.1 | 93.5 | 87.1 | 74.9 | 62.0 | 26.0 | 18.6 | 15.5 | 38.6 | 99.6 | 26.3 | 20.3 | 19.1 | 41.9 | 99.6 | 35.5 | 41.5 | 47.0 | 43.7 | 45.6 |
| GFY | 96.5 | 94.2 | 80.7 | 56.7 | 32.0 | 9.6 | 9.2 | 6.9 | 18.8 | 99.0 | 17.6 | 13.1 | 8.1 | 20.8 | 99.1 | 46.6 | 43.7 | 36.1 | 39.5 | 49.3 |
| GBY | 96.1 | 92.9 | 82.0 | 60.5 | 36.3 | 17.2 | 15.1 | 8.1 | 28.5 | 99.2 | 20.2 | 18.1 | 10.0 | 31.1 | 99.2 | 49.7 | 48.0 | 39.2 | 38.7 | 47.3 |
| DFX | 83.8 | 42.2 | 0.7 | 0.0 | 0.0 | N/A | N/A | N/A | N/A | N/A | N/A | N/A | N/A | N/A | N/A | 63.3 | 66.6 | 48.4 | 97.7 | 100.0 |
| DBX | 90.9 | 78.6 | 50.7 | 31.7 | 18.3 | N/A | N/A | N/A | N/A | N/A | N/A | N/A | N/A | N/A | N/A | 50.7 | 58.7 | 56.8 | 59.7 | 57.2 |
| DFZ | 83.9 | 52.1 | 2.9 | 0.0 | 0.0 | 6.6 | 3.8 | 2.2 | 3.9 | 60.6 | 6.9 | 4.2 | 2.4 | 3.3 | 60.4 | 57.5 | 62.3 | 54.8 | 66.7 | 99.8 |

Table E.6: PGD white-box zero-knowledge attack results on CIFAR plane-vs-frog binary classification.

| | acc. (adv) | | | | | TP marginal | | | | | TP logit | | | | | TP KL | | | | |
|---|---|---|---|---|---|---|---|---|---|---|---|---|---|---|---|---|---|---|---|---|
| $\epsilon$ | 0.01 | 0.02 | 0.05 | 0.10 | 0.20 | 0.01 | 0.02 | 0.05 | 0.10 | 0.20 | 0.01 | 0.02 | 0.05 | 0.10 | 0.20 | 0.01 | 0.02 | 0.05 | 0.10 | 0.20 |
| BNN | 97.9 | 86.7 | 19.7 | 1.0 | 0.0 | N/A | N/A | N/A | N/A | N/A | N/A | N/A | N/A | N/A | N/A | 41.7 | 59.4 | 55.7 | 68.4 | 98.9 |
| GFZ | 98.0 | 93.9 | 67.7 | 21.7 | 3.5 | 3.3 | 6.6 | 5.6 | 7.8 | 32.9 | 5.0 | 9.5 | 6.0 | 8.2 | 33.0 | 37.5 | 45.1 | 34.5 | 44.0 | 57.7 |
| GBZ | 94.6 | 93.7 | 83.9 | 67.4 | 52.8 | 19.8 | 17.9 | 12.8 | 14.3 | 43.1 | 22.8 | 19.5 | 17.4 | 17.0 | 44.3 | 32.1 | 39.3 | 40.2 | 37.1 | 33.2 |
| GFY | 98.4 | 95.0 | 67.9 | 25.8 | 4.1 | 4.2 | 6.7 | 6.4 | 7.8 | 35.7 | 3.1 | 8.2 | 7.6 | 7.8 | 33.8 | 31.2 | 43.6 | 32.9 | 39.6 | 53.3 |
| GBY | 96.4 | 92.9 | 67.3 | 25.7 | 6.9 | 11.1 | 8.4 | 7.9 | 11.6 | 41.4 | 13.1 | 11.5 | 9.2 | 12.2 | 40.3 | 43.0 | 43.0 | 33.6 | 39.1 | 52.2 |
| DFX | 82.7 | 35.7 | 0.3 | 0.0 | 0.0 | N/A | N/A | N/A | N/A | N/A | N/A | N/A | N/A | N/A | N/A | 64.1 | 64.7 | 69.6 | 100.0 | 100.0 |
| DBX | 83.3 | 34.6 | 4.2 | 0.7 | 0.0 | N/A | N/A | N/A | N/A | N/A | N/A | N/A | N/A | N/A | N/A | 59.8 | 59.5 | 65.1 | 78.7 | 93.4 |
| DFZ | 82.4 | 36.9 | 0.4 | 0.0 | 0.0 | 4.9 | 3.8 | 5.1 | 12.3 | 91.1 | 6.5 | 4.1 | 5.5 | 12.3 | 91.3 | 56.3 | 60.3 | 54.7 | 99.7 | 100.0 |

Table E.7: MIM white-box zero-knowledge attack results on CIFAR plane-vs-frog binary classification.

| | acc. (adv) | | | | | TP marginal | | | | | TP logit | | | | | TP KL | | | | |
|---|---|---|---|---|---|---|---|---|---|---|---|---|---|---|---|---|---|---|---|---|
| $\epsilon$ | 0.01 | 0.02 | 0.05 | 0.10 | 0.20 | 0.01 | 0.02 | 0.05 | 0.10 | 0.20 | 0.01 | 0.02 | 0.05 | 0.10 | 0.20 | 0.01 | 0.02 | 0.05 | 0.10 | 0.20 |
| BNN | 96.9 | 84.6 | 18.7 | 0.9 | 0.0 | N/A | N/A | N/A | N/A | N/A | N/A | N/A | N/A | N/A | N/A | 35.9 | 54.7 | 56.3 | 62.4 | 78.8 |
| GFZ | 96.9 | 91.5 | 58.1 | 15.2 | 1.9 | 6.5 | 6.2 | 6.4 | 10.6 | 86.1 | 10.0 | 8.3 | 6.6 | 11.3 | 84.5 | 31.0 | 39.0 | 32.1 | 47.2 | 77.1 |
| GBZ | 92.5 | 89.4 | 71.5 | 40.4 | 17.9 | 15.0 | 13.2 | 11.0 | 24.0 | 95.9 | 16.4 | 14.8 | 12.8 | 25.1 | 96.0 | 42.5 | 37.5 | 31.0 | 37.4 | 55.0 |
| GFY | 97.6 | 92.3 | 59.5 | 16.5 | 2.3 | 5.6 | 6.9 | 6.3 | 13.3 | 88.1 | 5.7 | 6.2 | 6.8 | 13.8 | 87.6 | 23.8 | 41.4 | 33.8 | 43.9 | 74.6 |
| GBY | 95.1 | 88.8 | 56.5 | 14.6 | 1.3 | 15.1 | 9.5 | 8.6 | 19.2 | 97.3 | 15.3 | 10.7 | 9.9 | 18.5 | 97.0 | 51.1 | 43.2 | 31.7 | 46.7 | 87.6 |
| DFX | 82.5 | 35.2 | 0.3 | 0.0 | 0.0 | N/A | N/A | N/A | N/A | N/A | N/A | N/A | N/A | N/A | N/A | 65.5 | 67.1 | 69.6 | 100.0 | 100.0 |
| DBX | 82.5 | 38.8 | 6.1 | 1.2 | 0.1 | N/A | N/A | N/A | N/A | N/A | N/A | N/A | N/A | N/A | N/A | 54.6 | 60.0 | 61.5 | 78.8 | 88.5 |
| DFZ | 81.5 | 36.1 | 0.4 | 0.0 | 0.0 | 5.4 | 3.9 | 5.0 | 13.4 | 96.9 | 6.2 | 4.1 | 5.5 | 13.8 | 96.8 | 61.1 | 58.5 | 55.3 | 99.9 | 100.0 |

Table E.8: CW white-box zero-knowledge attack results on CIFAR plane-vs-frog binary classification.

| | acc. (adv) | | | | | TP marginal | | | | | TP logit | | | | | TP KL | | | | |
|---|---|---|---|---|---|---|---|---|---|---|---|---|---|---|---|---|---|---|---|---|
| $\epsilon$ | 0.10 | 1.00 | 10.00 | 100.00 | 1000.00 | 0.10 | 1.00 | 10.00 | 100.00 | 1000.00 | 0.10 | 1.00 | 10.00 | 100.00 | 1000.00 | 0.10 | 1.00 | 10.00 | 100.00 | 1000.00 |
| BNN | 93.7 | 66.2 | 37.3 | 32.3 | 47.6 | N/A | N/A | N/A | N/A | N/A | N/A | N/A | N/A | N/A | N/A | 60.9 | 80.3 | 82.2 | 72.0 | 80.0 |
| GFZ | 99.5 | 95.6 | 76.5 | 77.9 | 78.3 | 0.0 | 4.5 | 2.6 | 3.0 | 3.4 | 0.0 | 4.6 | 3.0 | 3.6 | 4.2 | 29.2 | 30.4 | 25.6 | 27.1 | 29.0 |
| GBZ | 96.0 | 93.6 | 88.9 | 88.7 | 89.0 | 16.7 | 11.5 | 7.8 | 7.7 | 7.9 | 18.9 | 13.7 | 9.3 | 9.1 | 9.4 | 41.1 | 51.4 | 40.4 | 34.7 | 32.0 |
| GFY | 99.8 | 95.9 | 78.8 | 80.0 | 80.5 | 0.0 | 6.6 | 4.1 | 4.7 | 5.1 | 0.0 | 6.6 | 4.8 | 4.9 | 5.4 | 75.0 | 37.3 | 26.0 | 26.4 | 31.1 |
| GBY | 97.5 | 93.1 | 76.7 | 77.3 | 78.0 | 15.8 | 14.6 | 6.0 | 7.1 | 7.3 | 18.9 | 13.9 | 6.0 | 7.0 | 7.9 | 42.6 | 34.3 | 27.2 | 31.6 | 29.0 |
| DFX | 82.6 | 44.2 | 34.3 | 6.1 | 0.6 | N/A | N/A | N/A | N/A | N/A | N/A | N/A | N/A | N/A | N/A | 100.0 | 100.0 | 100.0 | 82.5 | 95.0 |
| DBX | 96.5 | 72.2 | 26.3 | 6.3 | 16.2 | N/A | N/A | N/A | N/A | N/A | N/A | N/A | N/A | N/A | N/A | 61.6 | 79.4 | 85.2 | 62.2 | 67.9 |
| DFZ | 94.5 | 72.3 | 29.9 | 6.3 | 17.9 | 7.3 | 5.8 | 4.3 | 4.1 | 4.2 | 14.1 | 6.9 | 5.0 | 4.4 | 4.4 | 82.6 | 97.3 | 97.1 | 83.0 | 80.2 |

Table E.9: Perfect knowledge attacks on CIFAR binary task. This is done using the PGD attack with $\epsilon = 0.1$. 'ZK' stands for zero knowledge, 'PK0' where you have knowledge of the $K$ samples but not of the detection system. 'PKM', 'PKL', and 'PKK' are attacks where you have knowledge of the K samples and of the marginal, logit and KL detection mechanisms respectively.

| | ZK | | | | PK0 | | | | PKM | | | | PKL | | | | PKK | | | |
|---|---|---|---|---|---|---|---|---|---|---|---|---|---|---|---|---|---|---|---|---|
| | acc. | TP marg. | TP logit | TP KL | acc. | TP marg. | TP logit | TP KL | acc. | TP marg. | TP logit | TP KL | acc. | TP marg. | TP logit | TP KL | acc. | TP marg. | TP logit | TP KL |
| DBX | 0.7 | N/A | N/A | 78.7 | 0.7 | N/A | N/A | 82.5 | N/A | N/A | N/A | N/A | N/A | N/A | N/A | N/A | 0.6 | N/A | N/A | 82.6 |
| GBZ | 67.4 | 14.3 | 17.0 | 37.1 | 57 | 40.6 | 45.1 | 49 | 58.4 | 9.6 | 18.5 | 46.9 | 58.5 | 1.8 | 7.8 | 47.8 | 59.9 | 25.5 | 27.6 | 42.4 |
| GBY | 25.7 | 11.6 | 12.2 | 39.1 | 28.5 | 22.0 | 22.6 | 49 | 29.5 | 5.6 | 7.6 | 45.9 | 29.2 | 0.8 | 1.8 | 45.4 | 27.9 | 15.6 | 15.9 | 45 |

Table E.10: Grey-box PGD attack results on MNIST.

| | substitute acc | | | | | victim acc | | | | | TP logit | | | | |
| --- | --- | --- | --- | --- | --- | --- | --- | --- | --- | --- | --- | --- | --- | --- | --- |
| $\epsilon$ | 0.10 | 0.20 | 0.30 | 0.40 | 0.50 | 0.10 | 0.20 | 0.30 | 0.40 | 0.50 | 0.10 | 0.20 | 0.30 | 0.40 | 0.50 |
| GFZ | 86.7 | 14.3 | 0.0 | 0.0 | 0.0 | 96.6 | 83.3 | 51.3 | 26.8 | 17.2 | 49.5 | 73.8 | 99.7 | 100.0 | 100.0 |
| GBZ | 81.2 | 9.1 | 0.0 | 0.0 | 0.0 | 93.5 | 80.1 | 55.3 | 35.0 | 25.2 | 46.7 | 73.3 | 99.6 | 100.0 | 100.0 |
| GFY | 84.8 | 6.4 | 0.0 | 0.0 | 0.0 | 96.7 | 82.3 | 55.2 | 33.5 | 24.0 | 58.7 | 88.8 | 100.0 | 100.0 | 100.0 |
| GBY | 86.7 | 15.0 | 0.0 | 0.0 | 0.0 | 95.8 | 81.6 | 51.0 | 27.5 | 18.1 | 50.1 | 78.5 | 99.9 | 100.0 | 100.0 |
| DFX | 74.2 | 5.2 | 0.5 | 0.0 | 0.0 | 91.7 | 57.7 | 19.6 | 4.3 | 1.4 | N/A | N/A | N/A | N/A | N/A |
| DBX | 80.6 | 5.1 | 0.0 | 0.0 | 0.0 | 93.2 | 59.2 | 22.5 | 10.9 | 8.2 | N/A | N/A | N/A | N/A | N/A |
| DFZ | 69.6 | 3.4 | 0.3 | 0.0 | 0.0 | 91.9 | 57.2 | 21.4 | 6.3 | 2.6 | 33.6 | 55.1 | 91.7 | 100.0 | 100.0 |

Table E.11: Grey-box MIM attack results on MNIST.

| | substitute acc | | | | | victim acc | | | | | TP logit | | | | |
| --- | --- | --- | --- | --- | --- | --- | --- | --- | --- | --- | --- | --- | --- | --- | --- |
| $\epsilon$ | 0.10 | 0.20 | 0.30 | 0.40 | 0.50 | 0.10 | 0.20 | 0.30 | 0.40 | 0.50 | 0.10 | 0.20 | 0.30 | 0.40 | 0.50 |
| GFZ | 87.2 | 16.1 | 0.0 | 0.0 | 0.0 | 96.5 | 82.4 | 42.8 | 14.2 | 3.7 | 50.6 | 82.5 | 100.0 | 100.0 | 100.0 |
| GBZ | 81.8 | 9.1 | 0.0 | 0.0 | 0.0 | 93.3 | 78.6 | 45.3 | 17.0 | 5.2 | 46.2 | 81.7 | 100.0 | 100.0 | 100.0 |
| GFY | 85.2 | 6.9 | 0.0 | 0.0 | 0.0 | 96.6 | 80.8 | 48.0 | 18.9 | 7.4 | 59.8 | 96.0 | 100.0 | 100.0 | 100.0 |
| GBY | 87.0 | 15.9 | 0.0 | 0.0 | 0.0 | 95.7 | 79.9 | 41.8 | 13.0 | 4.0 | 51.4 | 87.1 | 100.0 | 100.0 | 100.0 |
| DFX | 76.5 | 12.3 | 3.6 | 2.1 | 2.1 | 91.5 | 56.1 | 22.2 | 9.9 | 9.9 | N/A | N/A | N/A | N/A | N/A |
| DBX | 85.1 | 19.9 | 0.6 | 0.1 | 0.1 | 93.4 | 57.0 | 14.1 | 6.3 | 6.3 | N/A | N/A | N/A | N/A | N/A |
| DFZ | 70.9 | 3.2 | 0.0 | 0.0 | 0.0 | 91.8 | 54.6 | 16.4 | 2.7 | 0.3 | 34.5 | 63.8 | 98.9 | 100.0 | 100.0 |

Table E.12: Grey-box CW attack results on MNIST.

| | substitute acc | | | | | victim acc | | | | | TP logit | | | | |
| --- | --- | --- | --- | --- | --- | --- | --- | --- | --- | --- | --- | --- | --- | --- | --- |
| $\epsilon$ | 0.10 | 1.00 | 10.00 | 100.00 | 1000.00 | 0.10 | 1.00 | 10.00 | 100.00 | 1000.00 | 0.10 | 1.00 | 10.00 | 100.00 | 1000.00 |
| GFZ | 98.4 | 4.6 | 0.0 | 0.0 | 0.0 | 98.8 | 96.9 | 96.0 | 93.4 | 90.2 | 53.4 | 44.2 | 41.6 | 40.6 | 43.7 |
| GBZ | 98.4 | 63.2 | 0.0 | 0.0 | 0.0 | 97.4 | 95.4 | 92.9 | 88.6 | 84.5 | 45.9 | 39.7 | 33.0 | 31.2 | 38.6 |
| GFY | 98.1 | 0.9 | 0.0 | 0.0 | 0.0 | 99.0 | 97.6 | 97.0 | 95.7 | 94.0 | 61.0 | 55.2 | 49.2 | 49.5 | 52.7 |
| GBY | 98.3 | 5.3 | 0.0 | 0.0 | 0.0 | 98.7 | 96.8 | 96.0 | 93.8 | 91.2 | 54.2 | 41.5 | 37.6 | 38.7 | 43.6 |
| DFX | 85.1 | 0.0 | 0.0 | 0.0 | 0.0 | 97.6 | 93.3 | 91.2 | 90.4 | 89.8 | N/A | N/A | N/A | N/A | N/A |
| DBX | 97.3 | 46.0 | 0.4 | 0.0 | 0.0 | 98.7 | 93.9 | 88.4 | 85.5 | 76.3 | N/A | N/A | N/A | N/A | N/A |
| DFZ | 80.5 | 0.0 | 0.0 | 0.0 | 0.0 | 97.2 | 94.6 | 92.3 | 91.5 | 91.2 | 32.9 | 28.6 | 28.1 | 36.5 | 44.5 |

Table E.13: Grey-box PGD attack results on CIFAR plane-vs-frog binary classification.

| | substitute acc | | | | | victim acc | | | | | TP logit | | | | |
|---|---|---|---|---|---|---|---|---|---|---|---|---|---|---|---|
| $\epsilon$ | 0.01 | 0.02 | 0.05 | 0.10 | 0.20 | 0.01 | 0.02 | 0.05 | 0.10 | 0.20 | 0.01 | 0.02 | 0.05 | 0.10 | 0.20 |
| GFZ | 96.7 | 88.1 | 41.7 | 4.5 | 0.1 | 97.0 | 94.9 | 84.0 | 48.3 | 8.3 | 8.4 | 11.2 | 6.1 | 7.5 | 49.8 |
| GBZ | 92.9 | 83.1 | 37.8 | 5.5 | 0.0 | 95.3 | 94.8 | 91.8 | 83.1 | 65.6 | 15.5 | 12.8 | 10.9 | 19.9 | 79.8 |
| GFY | 96.9 | 88.1 | 33.7 | 3.3 | 0.1 | 96.9 | 95.9 | 85.3 | 54.2 | 12.7 | 2.6 | 2.0 | 5.0 | 8.5 | 61.2 |
| GBY | 95.2 | 85.6 | 32.4 | 2.8 | 0.1 | 96.8 | 95.3 | 86.4 | 59.8 | 18.5 | 21.8 | 14.3 | 10.5 | 12.5 | 75.2 |
| DFX | 95.7 | 80.8 | 10.1 | 0.2 | 0.0 | 99.1 | 96.7 | 79.4 | 28.3 | 0.9 | N/A | N/A | N/A | N/A | N/A |
| DBX | 95.7 | 80.5 | 30.5 | 1.7 | 0.0 | 99.5 | 98.2 | 91.1 | 65.1 | 21.8 | N/A | N/A | N/A | N/A | N/A |
| DFZ | 96.5 | 81.9 | 10.5 | 0.3 | 0.0 | 99.1 | 96.9 | 76.9 | 26.2 | 1.3 | 4.5 | 16.5 | 6.1 | 8.5 | 59.6 |

Table E.14: Grey-box MIM attack results on CIFAR plane-vs-frog binary classification.

| | substitute acc | | | | | victim acc | | | | | TP logit | | | | |
|---|---|---|---|---|---|---|---|---|---|---|---|---|---|---|---|
| $\epsilon$ | 0.01 | 0.02 | 0.05 | 0.10 | 0.20 | 0.01 | 0.02 | 0.05 | 0.10 | 0.20 | 0.01 | 0.02 | 0.05 | 0.10 | 0.20 |
| GFZ | 96.3 | 87.9 | 42.1 | 4.7 | 0.1 | 97.1 | 94.8 | 83.0 | 44.8 | 4.9 | 8.7 | 11.1 | 5.8 | 8.5 | 75.1 |
| GBZ | 92.9 | 82.5 | 36.9 | 4.9 | 0.0 | 95.5 | 94.7 | 91.4 | 81.7 | 49.5 | 14.8 | 12.6 | 13.9 | 21.9 | 97.8 |
| GFY | 96.8 | 87.9 | 34.1 | 3.5 | 0.1 | 96.9 | 95.6 | 84.6 | 49.1 | 6.9 | 2.6 | 2.0 | 6.7 | 10.3 | 81.1 |
| GBY | 95.1 | 85.5 | 32.0 | 2.6 | 0.1 | 96.8 | 95.5 | 85.7 | 56.2 | 10.9 | 21.8 | 14.8 | 9.2 | 13.4 | 97.6 |
| DFX | 95.5 | 80.5 | 10.5 | 0.2 | 0.0 | 99.1 | 96.5 | 77.3 | 24.0 | 0.2 | N/A | N/A | N/A | N/A | N/A |
| DBX | 95.6 | 81.1 | 34.5 | 2.2 | 0.0 | 99.5 | 98.3 | 89.3 | 58.9 | 9.9 | N/A | N/A | N/A | N/A | N/A |
| DFZ | 96.3 | 81.5 | 10.7 | 0.5 | 0.0 | 99.1 | 96.9 | 74.9 | 21.9 | 0.3 | 4.5 | 16.6 | 6.4 | 9.9 | 80.2 |

Table E.15: Grey-box CW attack results on CIFAR plane-vs-frog binary classification.

| | substitute acc | | | | | victim acc | | | | | TP logit | | | | |
|---|---|---|---|---|---|---|---|---|---|---|---|---|---|---|---|
| $\epsilon$ | 0.10 | 1.00 | 10.00 | 100.00 | 1000.00 | 0.10 | 1.00 | 10.00 | 100.00 | 1000.00 | 0.10 | 1.00 | 10.00 | 100.00 | 1000.00 |
| GFZ | 98.7 | 61.9 | 0.0 | 0.0 | 0.0 | 98.6 | 95.8 | 94.3 | 93.8 | 93.2 | 11.7 | 10.2 | 10.9 | 9.9 | 8.5 |
| GBZ | 96.3 | 69.8 | 3.3 | 0.0 | 0.0 | 95.9 | 95.7 | 94.7 | 94.7 | 94.5 | 13.5 | 12.7 | 12.3 | 13.7 | 14.4 |
| GFY | 98.4 | 54.4 | 0.0 | 0.0 | 0.0 | 98.1 | 96.1 | 95.3 | 95.1 | 94.3 | 0.0 | 2.2 | 3.7 | 4.6 | 7.6 |
| GBY | 97.6 | 52.9 | 2.2 | 0.0 | 0.0 | 97.7 | 95.6 | 94.5 | 94.3 | 93.7 | 25.8 | 15.7 | 14.2 | 13.7 | 18.0 |
| DFX | 64.4 | 0.0 | 0.0 | 0.0 | 0.0 | 98.5 | 97.2 | 97.1 | 96.7 | 95.5 | N/A | N/A | N/A | N/A | N/A |
| DBX | 91.3 | 56.2 | 0.3 | 0.0 | 0.0 | 99.7 | 99.0 | 99.0 | 98.9 | 98.1 | N/A | N/A | N/A | N/A | N/A |
| DFZ | 80.3 | 0.0 | 0.0 | 0.0 | 0.0 | 99.2 | 97.9 | 97.5 | 97.3 | 95.4 | 0.0 | 15.7 | 14.6 | 8.8 | 6.1 |

Table E.16: Black-box PGD attack results on MNIST.

| $\epsilon$ | substitute acc | | | | | victim acc | | | | | TP logit | | | | |
|---|---|---|---|---|---|---|---|---|---|---|---|---|---|---|---|
| | 0.10 | 0.20 | 0.30 | 0.40 | 0.50 | 0.10 | 0.20 | 0.30 | 0.40 | 0.50 | 0.10 | 0.20 | 0.30 | 0.40 | 0.50 |
| GFZ | 44.2 | 0.7 | 0.0 | 0.0 | 0.0 | 97.3 | 88.4 | 62.7 | 33.8 | 21.8 | 48.7 | 61.0 | 92.3 | 99.8 | 100.0 |
| GBZ | 7.4 | 0.0 | 0.0 | 0.0 | 0.0 | 94.8 | 87.1 | 70.7 | 52.9 | 41.3 | 51.0 | 68.6 | 97.7 | 100.0 | 100.0 |
| GFY | 49.4 | 0.8 | 0.0 | 0.0 | 0.0 | 97.4 | 86.5 | 58.1 | 31.9 | 21.0 | 52.5 | 70.8 | 97.4 | 99.8 | 100.0 |
| GBY | 21.7 | 0.0 | 0.0 | 0.0 | 0.0 | 96.9 | 89.3 | 70.0 | 49.7 | 37.8 | 51.2 | 70.8 | 98.2 | 100.0 | 100.0 |
| DFX | 49.2 | 1.3 | 0.0 | 0.0 | 0.0 | 91.4 | 52.1 | 13.9 | 2.2 | 0.7 | N/A | N/A | N/A | N/A | N/A |
| DBX | 43.1 | 0.7 | 0.0 | 0.0 | 0.0 | 95.0 | 67.0 | 26.2 | 9.7 | 6.5 | N/A | N/A | N/A | N/A | N/A |
| DFZ | 53.8 | 1.7 | 0.0 | 0.0 | 0.0 | 92.9 | 56.3 | 15.2 | 3.4 | 1.5 | 34.7 | 52.8 | 94.0 | 99.9 | 100.0 |

Table E.17: Black-box MIM attack results on MNIST.

| $\epsilon$ | substitute acc | | | | | victim acc | | | | | TP logit | | | | |
|---|---|---|---|---|---|---|---|---|---|---|---|---|---|---|---|
| | 0.10 | 0.20 | 0.30 | 0.40 | 0.50 | 0.10 | 0.20 | 0.30 | 0.40 | 0.50 | 0.10 | 0.20 | 0.30 | 0.40 | 0.50 |
| GFZ | 45.5 | 2.8 | 0.0 | 0.0 | 0.0 | 97.2 | 86.6 | 50.3 | 17.0 | 17.0 | 51.1 | 68.0 | 97.5 | 99.5 | 99.5 |
| GBZ | 8.4 | 0.0 | 0.0 | 0.0 | 0.0 | 94.7 | 85.9 | 63.5 | 35.2 | 17.1 | 51.2 | 74.4 | 99.0 | 100.0 | 100.0 |
| GFY | 53.1 | 2.5 | 0.3 | 0.1 | 0.1 | 97.2 | 83.7 | 49.2 | 20.1 | 20.1 | 55.1 | 80.6 | 99.0 | 99.9 | 99.9 |
| GBY | 24.5 | 0.0 | 0.0 | 0.0 | 0.0 | 96.8 | 88.3 | 62.8 | 31.7 | 13.3 | 49.3 | 77.6 | 99.9 | 100.0 | 100.0 |
| DFX | 51.3 | 2.3 | 0.1 | 0.0 | 0.0 | 91.4 | 51.7 | 13.3 | 1.9 | 1.9 | N/A | N/A | N/A | N/A | N/A |
| DBX | 47.8 | 1.6 | 0.1 | 0.0 | 0.0 | 94.8 | 67.0 | 23.2 | 7.6 | 7.6 | N/A | N/A | N/A | N/A | N/A |
| DFZ | 56.1 | 2.9 | 0.1 | 0.0 | 0.0 | 92.7 | 56.0 | 13.1 | 2.8 | 2.8 | 35.2 | 59.9 | 97.6 | 100.0 | 100.0 |

Table E.18: Black-box CW attack results on MNIST.

| $\epsilon$ | substitute acc | | | | | victim acc | | | | | TP logit | | | | |
|---|---|---|---|---|---|---|---|---|---|---|---|---|---|---|---|
| | 0.10 | 1.00 | 10.00 | 100.00 | 1000.00 | 0.10 | 1.00 | 10.00 | 100.00 | 1000.00 | 0.10 | 1.00 | 10.00 | 100.00 | 1000.00 |
| GFZ | 65.2 | 0.3 | 0.0 | 0.0 | 0.0 | 98.8 | 98.7 | 97.4 | 94.3 | 92.0 | 52.3 | 50.8 | 39.6 | 44.7 | 55.3 |
| GBZ | 76.6 | 0.4 | 0.0 | 0.0 | 0.0 | 97.3 | 97.2 | 95.9 | 92.7 | 90.4 | 45.7 | 46.1 | 41.5 | 40.3 | 45.7 |
| GFY | 88.4 | 3.8 | 0.0 | 0.0 | 0.0 | 99.0 | 98.9 | 98.0 | 94.8 | 92.0 | 60.0 | 59.2 | 51.9 | 47.6 | 57.2 |
| GBY | 76.0 | 1.3 | 0.0 | 0.0 | 0.0 | 98.7 | 98.6 | 97.3 | 95.1 | 93.5 | 53.4 | 52.2 | 41.8 | 39.7 | 46.7 |
| DFX | 82.4 | 0.0 | 0.0 | 0.0 | 0.0 | 98.8 | 96.8 | 92.4 | 86.2 | 84.9 | N/A | N/A | N/A | N/A | N/A |
| DBX | 82.5 | 0.6 | 0.0 | 0.0 | 0.0 | 98.9 | 98.4 | 95.5 | 85.2 | 83.1 | N/A | N/A | N/A | N/A | N/A |
| DFZ | 86.5 | 0.2 | 0.0 | 0.0 | 0.0 | 98.8 | 94.5 | 89.3 | 81.5 | 79.3 | 42.9 | 25.5 | 22.3 | 27.7 | 38.2 |

Table E.19: Black-box PGD attack results on CIFAR plane-vs-frog binary classification.

| $\epsilon$ | substitute acc | | | | | victim acc | | | | | TP logit | | | | |
|---|---|---|---|---|---|---|---|---|---|---|---|---|---|---|---|
| | 0.01 | 0.02 | 0.05 | 0.10 | 0.20 | 0.01 | 0.02 | 0.05 | 0.10 | 0.20 | 0.01 | 0.02 | 0.05 | 0.10 | 0.20 |
| GFZ | 95.5 | 89.9 | 45.5 | 6.8 | 0.1 | 97.6 | 95.6 | 88.7 | 68.8 | 29.7 | 6.2 | 6.7 | 6.0 | 10.8 | 71.8 |
| GBZ | 92.0 | 84.1 | 33.8 | 3.5 | 0.0 | 95.6 | 94.9 | 92.7 | 85.0 | 65.9 | 16.7 | 14.6 | 12.8 | 17.9 | 76.8 |
| GFY | 95.3 | 89.1 | 38.7 | 2.5 | 0.0 | 97.3 | 96.6 | 90.0 | 72.9 | 35.5 | 0.0 | 2.3 | 6.8 | 10.4 | 72.3 |
| GBY | 91.8 | 80.5 | 29.7 | 6.1 | 0.2 | 97.2 | 95.9 | 90.2 | 70.8 | 30.3 | 21.8 | 16.4 | 9.6 | 13.6 | 77.3 |
| DFX | 89.1 | 74.5 | 19.1 | 0.9 | 0.0 | 99.7 | 98.7 | 92.6 | 64.8 | 9.9 | N/A | N/A | N/A | N/A | N/A |
| DBX | 85.5 | 76.4 | 42.5 | 4.5 | 0.0 | 99.6 | 99.0 | 94.8 | 77.5 | 24.5 | N/A | N/A | N/A | N/A | N/A |
| DFZ | 85.5 | 73.5 | 21.1 | 1.5 | 0.0 | 99.5 | 99.2 | 93.8 | 70.1 | 15.1 | 0.0 | 5.0 | 8.9 | 15.9 | 85.1 |

Table E.20: Black-box MIM attack results on CIFAR plane-vs-frog binary classification.

| $\epsilon$ | substitute acc | | | | | victim acc | | | | | TP logit | | | | |
|---|---|---|---|---|---|---|---|---|---|---|---|---|---|---|---|
| | 0.01 | 0.02 | 0.05 | 0.10 | 0.20 | 0.01 | 0.02 | 0.05 | 0.10 | 0.20 | 0.01 | 0.02 | 0.05 | 0.10 | 0.20 |
| GFZ | 95.5 | 89.7 | 46.4 | 7.7 | 0.2 | 97.5 | 95.7 | 88.5 | 66.5 | 20.5 | 6.2 | 6.7 | 5.9 | 12.7 | 82.7 |
| GBZ | 91.8 | 84.0 | 33.1 | 3.4 | 0.0 | 95.6 | 94.9 | 92.3 | 85.1 | 60.7 | 16.7 | 15.5 | 14.2 | 21.6 | 97.9 |
| GFY | 95.3 | 88.9 | 40.5 | 3.1 | 0.0 | 97.3 | 96.7 | 89.5 | 70.6 | 31.9 | 0.0 | 2.4 | 7.6 | 12.5 | 88.0 |
| GBY | 91.7 | 80.2 | 29.5 | 5.9 | 0.1 | 97.3 | 95.9 | 89.6 | 69.0 | 26.1 | 22.2 | 16.1 | 9.7 | 14.0 | 95.9 |
| DFX | 89.0 | 74.5 | 19.4 | 0.9 | 0.0 | 99.6 | 98.7 | 92.1 | 63.0 | 10.9 | N/A | N/A | N/A | N/A | N/A |
| DBX | 85.5 | 76.3 | 42.7 | 4.5 | 0.0 | 99.6 | 99.0 | 95.1 | 76.5 | 27.2 | N/A | N/A | N/A | N/A | N/A |
| DFZ | 85.3 | 73.4 | 21.4 | 1.6 | 0.0 | 99.6 | 99.1 | 92.9 | 67.7 | 14.9 | 0.0 | 4.5 | 7.5 | 14.2 | 91.9 |

Table E.21: Black-box CW attack results on CIFAR plane-vs-frog binary classification.

| $\epsilon$ | substitute acc | | | | | victim acc | | | | | TP logit | | | | |
|---|---|---|---|---|---|---|---|---|---|---|---|---|---|---|---|
| | 0.10 | 1.00 | 10.00 | 100.00 | 1000.00 | 0.10 | 1.00 | 10.00 | 100.00 | 1000.00 | 0.10 | 1.00 | 10.00 | 100.00 | 1000.00 |
| GFZ | 96.3 | 41.5 | 0.5 | 0.0 | 0.0 | 98.7 | 94.7 | 93.4 | 93.3 | 92.3 | 13.3 | 8.6 | 7.7 | 7.7 | 8.9 |
| GBZ | 94.9 | 73.3 | 1.4 | 0.0 | 0.0 | 95.9 | 95.3 | 94.6 | 94.5 | 94.5 | 13.5 | 11.8 | 11.8 | 13.2 | 13.1 |
| GFY | 96.4 | 62.0 | 1.3 | 0.0 | 0.0 | 98.1 | 96.7 | 94.8 | 94.7 | 94.1 | 0.0 | 2.8 | 6.5 | 6.1 | 6.8 |
| GBY | 93.5 | 33.6 | 1.7 | 0.0 | 0.0 | 97.7 | 95.9 | 95.4 | 95.4 | 95.0 | 25.8 | 16.4 | 15.9 | 18.6 | 19.8 |
| DFX | 90.3 | 10.9 | 0.0 | 0.0 | 0.0 | 99.9 | 98.8 | 98.5 | 98.5 | 97.9 | N/A | N/A | N/A | N/A | N/A |
| DBX | 87.2 | 31.8 | 0.0 | 0.0 | 0.0 | 99.9 | 97.1 | 94.5 | 93.9 | 94.7 | N/A | N/A | N/A | N/A | N/A |
| DFZ | 84.9 | 22.7 | 0.0 | 0.0 | 0.0 | 99.9 | 99.0 | 98.3 | 98.3 | 97.8 | 0.0 | 3.8 | 2.3 | 2.3 | 11.0 |

Table E.22: FGSM white-box zero-knowledge attack results on CIFAR-10.

| | acc. (adv) | | | | | TP marginal | | | | | TP logit | | | | | TP KL | | | | |
|---|---|---|---|---|---|---|---|---|---|---|---|---|---|---|---|---|---|---|---|---|
| $\epsilon$ | 0.01 | 0.02 | 0.05 | 0.10 | 0.20 | 0.01 | 0.02 | 0.05 | 0.10 | 0.20 | 0.01 | 0.02 | 0.05 | 0.10 | 0.20 | 0.01 | 0.02 | 0.05 | 0.10 | 0.20 |
| VGG16 | 44.5 | 25.8 | 16.6 | 11.9 | 10.1 | N/A | N/A | N/A | N/A | N/A | N/A | N/A | N/A | N/A | N/A | 95.1 | 95.7 | 97.0 | 97.5 | 97.1 |
| GBZ-FC1 | 63.7 | 49.7 | 34.3 | 18.9 | 12.7 | 60.0 | 62.4 | 80.6 | 90.9 | 94.1 | 63.7 | 65.0 | 80.6 | 95.0 | 98.4 | 90.1 | 90.8 | 92.3 | 92.6 | 92.6 |
| GBY-FC1 | 64.9 | 50.0 | 36.1 | 19.5 | 11.3 | 52.2 | 59.0 | 80.4 | 90.1 | 92.5 | 54.6 | 61.9 | 80.3 | 95.6 | 98.4 | 90.7 | 91.3 | 92.1 | 92.4 | 90.3 |
| DBX-FC1 | 57.3 | 45.7 | 31.8 | 16.3 | 10.8 | N/A | N/A | N/A | N/A | N/A | N/A | N/A | N/A | N/A | N/A | 92.8 | 93.2 | 94.0 | 94.3 | 92.8 |
| GBZ-CONV9 | 81.6 | 74.1 | 58.5 | 48.9 | 45.8 | 13.7 | 17.0 | 28.4 | 76.0 | 90.2 | 18.3 | 20.6 | 31.7 | 64.3 | 71.6 | 91.7 | 91.5 | 91.7 | 93.3 | 86.4 |
| GBY-CONV9 | 75.8 | 66.0 | 39.7 | 22.5 | 17.5 | 14.0 | 18.2 | 29.4 | 74.4 | 88.9 | 17.7 | 20.8 | 25.8 | 55.2 | 68.9 | 89.6 | 89.9 | 90.0 | 89.1 | 89.2 |
| DBX-CONV9 | 59.0 | 45.7 | 31.8 | 17.8 | 11.3 | N/A | N/A | N/A | N/A | N/A | N/A | N/A | N/A | N/A | N/A | 93.7 | 94.1 | 94.9 | 95.2 | 93.7 |

Table E.23: PGD white-box zero-knowledge attack results on CIFAR-10.

| | acc. (adv) | | | | | TP marginal | | | | | TP logit | | | | | TP KL | | | | |
|---|---|---|---|---|---|---|---|---|---|---|---|---|---|---|---|---|---|---|---|---|
| $\epsilon$ | 0.01 | 0.02 | 0.05 | 0.10 | 0.20 | 0.01 | 0.02 | 0.05 | 0.10 | 0.20 | 0.01 | 0.02 | 0.05 | 0.10 | 0.20 | 0.01 | 0.02 | 0.05 | 0.10 | 0.20 |
| VGG16 | 18.8 | 0.6 | 0.0 | 0.0 | 0.0 | N/A | N/A | N/A | N/A | N/A | N/A | N/A | N/A | N/A | N/A | 93.9 | 91.8 | 67.4 | 69.9 | 66.4 |
| GBZ-FC1 | 37.7 | 6.1 | 0.1 | 0.0 | 0.0 | 30.7 | 22.2 | 76.0 | 92.0 | 94.1 | 33.5 | 24.9 | 77.0 | 91.7 | 90.2 | 90.8 | 93.7 | 97.3 | 98.5 | 95.9 |
| GBY-FC1 | 31.2 | 8.5 | 2.6 | 1.5 | 0.6 | 32.0 | 20.3 | 82.3 | 95.1 | 97.6 | 36.6 | 29.5 | 84.1 | 93.7 | 90.4 | 90.3 | 94.1 | 97.7 | 97.3 | 96.9 |
| DBX-FC1 | 23.4 | 1.7 | 0.0 | 0.0 | 0.0 | N/A | N/A | N/A | N/A | N/A | N/A | N/A | N/A | N/A | N/A | 91.9 | 91.0 | 5.6 | 7.4 | 5.9 |
| GBZ-CONV9 | 61.1 | 38.3 | 24.2 | 10.2 | 2.3 | 19.2 | 60.3 | 96.7 | 98.9 | 99.6 | 20.1 | 60.3 | 96.8 | 98.3 | 98.7 | 90.3 | 93.1 | 97.7 | 98.7 | 99.6 |
| GBY-CONV9 | 66.5 | 26.6 | 3.2 | 0.4 | 0.0 | 17.0 | 55.6 | 95.7 | 99.2 | 99.6 | 17.8 | 52.7 | 95.3 | 98.6 | 97.6 | 89.6 | 91.8 | 98.7 | 99.4 | 96.9 |
| DBX-CONV9 | 24.0 | 3.0 | 0.3 | 0.1 | 0.0 | N/A | N/A | N/A | N/A | N/A | N/A | N/A | N/A | N/A | N/A | 91.3 | 93.1 | 99.3 | 99.6 | 99.2 |

Table E.24: MIM white-box zero-knowledge attack results on CIFAR-10.

| | acc. (adv) | | | | | TP marginal | | | | | TP logit | | | | | TP KL | | | | |
|---|---|---|---|---|---|---|---|---|---|---|---|---|---|---|---|---|---|---|---|---|
| $\epsilon$ | 0.01 | 0.02 | 0.05 | 0.10 | 0.20 | 0.01 | 0.02 | 0.05 | 0.10 | 0.20 | 0.01 | 0.02 | 0.05 | 0.10 | 0.20 | 0.01 | 0.02 | 0.05 | 0.10 | 0.20 |
| VGG16 | 13.1 | 0.4 | 0.0 | 0.0 | 0.0 | N/A | N/A | N/A | N/A | N/A | N/A | N/A | N/A | N/A | N/A | 92.2 | 92.4 | 72.3 | 77.1 | 79.0 |
| GBZ-FC1 | 24.5 | 2.7 | 0.1 | 0.0 | 0.0 | 34.8 | 40.9 | 88.5 | 95.8 | 96.0 | 38.8 | 44.0 | 90.4 | 96.4 | 96.6 | 91.3 | 95.0 | 95.7 | 97.9 | 96.8 |
| GBY-FC1 | 33.0 | 19.6 | 11.8 | 7.2 | 4.9 | 29.1 | 38.7 | 95.7 | 99.1 | 99.3 | 38.4 | 49.2 | 96.3 | 99.3 | 99.4 | 90.7 | 95.0 | 97.8 | 97.9 | 97.9 |
| DBX-FC1 | 17.3 | 0.9 | 0.0 | 0.0 | 0.0 | N/A | N/A | N/A | N/A | N/A | N/A | N/A | N/A | N/A | N/A | 91.1 | 89.6 | 9.6 | 14.5 | 16.0 |
| GBZ-CONV9 | 57.6 | 43.5 | 29.1 | 24.1 | 22.6 | 27.4 | 78.5 | 99.4 | 99.6 | 99.7 | 29.1 | 78.3 | 99.6 | 99.7 | 99.8 | 90.2 | 94.4 | 97.5 | 98.0 | 98.2 |
| GBY-CONV9 | 39.1 | 18.1 | 4.3 | 1.9 | 1.5 | 25.6 | 70.4 | 98.6 | 99.6 | 99.8 | 28.2 | 71.1 | 98.6 | 99.7 | 99.8 | 89.6 | 94.5 | 99.4 | 99.8 | 99.7 |
| DBX-CONV9 | 19.2 | 3.2 | 0.5 | 0.4 | 0.4 | N/A | N/A | N/A | N/A | N/A | N/A | N/A | N/A | N/A | N/A | 92.1 | 96.7 | 99.9 | 100.0 | 100.0 |

Table E.25: FGSM white-box zero-knowledge attack results on MNIST (with varied bottleneck layer sizes).

| | acc. (adv) | | | | | TP marginal | | | | | TP logit | | | | | TP KL | | | | |
|---|---|---|---|---|---|---|---|---|---|---|---|---|---|---|---|---|---|---|---|---|
| $\epsilon$ | 0.10 | 0.20 | 0.30 | 0.40 | 0.50 | 0.10 | 0.20 | 0.30 | 0.40 | 0.50 | 0.10 | 0.20 | 0.30 | 0.40 | 0.50 | 0.10 | 0.20 | 0.30 | 0.40 | 0.50 |
| DBX-16 | 92.6 | 85.6 | 76.4 | 64.7 | 52.3 | N/A | N/A | N/A | N/A | N/A | N/A | N/A | N/A | N/A | N/A | 90.3 | 92.4 | 92.3 | 92.7 | 94.2 |
| DBX-32 | 92.6 | 84.4 | 71.1 | 57.2 | 46.0 | N/A | N/A | N/A | N/A | N/A | N/A | N/A | N/A | N/A | N/A | 92.2 | 92.3 | 92.8 | 94.9 | 95.0 |
| DBX-64 | 91.6 | 77.8 | 58.1 | 44.5 | 36.0 | N/A | N/A | N/A | N/A | N/A | N/A | N/A | N/A | N/A | N/A | 92.6 | 93.6 | 95.2 | 96.3 | 97.0 |
| DBX-128 | 87.8 | 47.6 | 20.1 | 12.7 | 10.3 | N/A | N/A | N/A | N/A | N/A | N/A | N/A | N/A | N/A | N/A | 94.4 | 97.6 | 97.8 | 97.1 | 96.4 |
| GBZ-16 | 92.0 | 75.8 | 52.5 | 34.0 | 21.7 | 28.7 | 71.5 | 99.9 | 100.0 | 100.0 | 49.2 | 86.3 | 99.9 | 100.0 | 100.0 | 90.8 | 90.6 | 92.0 | 91.9 | 90.9 |
| GBZ-32 | 91.1 | 74.4 | 52.4 | 33.4 | 21.6 | 26.5 | 73.5 | 99.7 | 100.0 | 100.0 | 49.4 | 87.2 | 99.9 | 100.0 | 100.0 | 91.0 | 90.6 | 91.8 | 91.4 | 91.6 |
| GBZ-64 | 92.5 | 80.3 | 62.0 | 42.4 | 27.2 | 37.0 | 81.7 | 100.0 | 100.0 | 100.0 | 57.6 | 93.5 | 100.0 | 100.0 | 100.0 | 91.6 | 90.9 | 90.3 | 91.1 | 91.5 |
| GBZ-128 | 90.8 | 76.8 | 57.5 | 38.9 | 24.8 | 26.6 | 63.0 | 99.6 | 100.0 | 100.0 | 44.9 | 78.7 | 99.8 | 100.0 | 100.0 | 87.9 | 90.1 | 90.7 | 91.1 | 90.6 |
| GBY-16 | 94.2 | 77.7 | 49.4 | 23.9 | 12.1 | 41.9 | 84.1 | 100.0 | 100.0 | 100.0 | 52.9 | 91.9 | 100.0 | 100.0 | 100.0 | 86.6 | 92.0 | 93.0 | 93.1 | 92.6 |
| GBY-32 | 94.5 | 76.9 | 45.4 | 20.0 | 9.6 | 41.4 | 83.3 | 100.0 | 100.0 | 100.0 | 56.3 | 92.8 | 100.0 | 100.0 | 100.0 | 89.0 | 91.6 | 93.3 | 93.2 | 93.0 |
| GBY-64 | 93.6 | 76.4 | 47.5 | 22.3 | 10.7 | 41.9 | 84.5 | 100.0 | 100.0 | 100.0 | 57.7 | 92.5 | 100.0 | 100.0 | 100.0 | 89.3 | 92.7 | 92.8 | 92.9 | 92.9 |
| GBY-128 | 93.0 | 72.9 | 42.2 | 18.5 | 9.0 | 37.4 | 71.6 | 100.0 | 100.0 | 100.0 | 50.1 | 81.9 | 99.9 | 100.0 | 100.0 | 90.7 | 91.4 | 91.8 | 91.4 | 92.0 |

Table E.26: PGD white-box zero-knowledge attack results on MNIST (with varied bottleneck layer sizes).

| | acc. (adv) | | | | | TP marginal | | | | | TP logit | | | | | TP KL | | | | |
|---|---|---|---|---|---|---|---|---|---|---|---|---|---|---|---|---|---|---|---|---|
| $\epsilon$ | 0.10 | 0.20 | 0.30 | 0.40 | 0.50 | 0.10 | 0.20 | 0.30 | 0.40 | 0.50 | 0.10 | 0.20 | 0.30 | 0.40 | 0.50 | 0.10 | 0.20 | 0.30 | 0.40 | 0.50 |
| DBX-16 | 69.8 | 36.6 | 16.0 | 5.9 | 1.8 | N/A | N/A | N/A | N/A | N/A | N/A | N/A | N/A | N/A | N/A | 91.2 | 91.9 | 92.8 | 93.3 | 93.7 |
| DBX-32 | 63.5 | 26.8 | 12.5 | 6.2 | 2.7 | N/A | N/A | N/A | N/A | N/A | N/A | N/A | N/A | N/A | N/A | 91.5 | 92.1 | 92.7 | 92.9 | 93.5 |
| DBX-64 | 58.0 | 18.3 | 6.0 | 1.3 | 0.2 | N/A | N/A | N/A | N/A | N/A | N/A | N/A | N/A | N/A | N/A | 92.7 | 94.0 | 94.1 | 93.7 | 84.8 |
| DBX-128 | 42.3 | 3.0 | 1.1 | 0.3 | 0.2 | N/A | N/A | N/A | N/A | N/A | N/A | N/A | N/A | N/A | N/A | 97.1 | 97.1 | 98.4 | 98.4 | 98.6 |
| GBZ-16 | 88.3 | 62.8 | 37.4 | 23.1 | 15.5 | 27.1 | 35.8 | 71.9 | 98.9 | 100.0 | 46.3 | 53.0 | 82.3 | 99.3 | 100.0 | 89.6 | 91.7 | 92.3 | 91.4 | 91.8 |
| GBZ-32 | 87.1 | 61.9 | 40.2 | 28.0 | 20.4 | 24.0 | 39.1 | 76.7 | 98.9 | 100.0 | 41.5 | 56.4 | 85.4 | 99.7 | 100.0 | 91.3 | 91.0 | 91.1 | 91.3 | 91.1 |
| GBZ-64 | 85.1 | 57.4 | 32.5 | 19.3 | 11.6 | 33.7 | 50.3 | 84.4 | 99.7 | 100.0 | 52.2 | 66.3 | 91.5 | 99.8 | 100.0 | 90.0 | 91.5 | 91.5 | 91.7 | 91.8 |
| GBZ-128 | 84.4 | 57.5 | 36.8 | 25.2 | 16.8 | 25.4 | 35.4 | 66.1 | 96.6 | 100.0 | 41.5 | 50.5 | 76.4 | 97.7 | 100.0 | 88.6 | 90.4 | 90.4 | 90.0 | 90.6 |
| GBY-16 | 90.4 | 49.7 | 15.0 | 4.2 | 1.6 | 42.6 | 64.7 | 89.9 | 99.3 | 100.0 | 51.1 | 70.8 | 92.8 | 99.5 | 100.0 | 91.9 | 93.1 | 91.9 | 90.7 | 91.0 |
| GBY-32 | 88.4 | 39.8 | 9.5 | 1.8 | 0.6 | 45.6 | 74.3 | 92.8 | 99.5 | 100.0 | 56.8 | 78.2 | 94.8 | 99.8 | 100.0 | 90.8 | 92.2 | 91.7 | 91.0 | 90.6 |
| GBY-64 | 86.7 | 35.9 | 9.0 | 1.7 | 0.4 | 45.3 | 76.1 | 94.6 | 99.7 | 100.0 | 56.9 | 79.3 | 95.6 | 99.9 | 100.0 | 90.1 | 92.1 | 91.6 | 91.7 | 91.4 |
| GBY-128 | 83.3 | 35.1 | 8.7 | 2.3 | 0.8 | 39.1 | 63.9 | 86.7 | 98.3 | 100.0 | 51.7 | 69.1 | 89.2 | 98.9 | 100.0 | 89.8 | 90.7 | 90.4 | 89.7 | 89.5 |

Table E.27: MIM white-box zero-knowledge attack results on MNIST (with varied bottleneck layer sizes).

| | acc. (adv) | | | | | TP marginal | | | | | TP logit | | | | | TP KL | | | | |
|---|---|---|---|---|---|---|---|---|---|---|---|---|---|---|---|---|---|---|---|---|
| $\epsilon$ | 0.10 | 0.20 | 0.30 | 0.40 | 0.50 | 0.10 | 0.20 | 0.30 | 0.40 | 0.50 | 0.10 | 0.20 | 0.30 | 0.40 | 0.50 | 0.10 | 0.20 | 0.30 | 0.40 | 0.50 |
| DBX-16 | 72.2 | 37.1 | 20.7 | 14.7 | 11.7 | N/A | N/A | N/A | N/A | N/A | N/A | N/A | N/A | N/A | N/A | 91.8 | 92.3 | 92.8 | 93.4 | 93.9 |
| DBX-32 | 66.7 | 27.6 | 16.8 | 12.5 | 10.6 | N/A | N/A | N/A | N/A | N/A | N/A | N/A | N/A | N/A | N/A | 91.8 | 92.5 | 92.4 | 93.2 | 93.5 |
| DBX-64 | 66.7 | 28.7 | 19.7 | 17.2 | 17.2 | N/A | N/A | N/A | N/A | N/A | N/A | N/A | N/A | N/A | N/A | 93.5 | 94.3 | 94.6 | 94.7 | 94.7 |
| DBX-128 | 42.3 | 1.3 | 0.2 | 0.1 | 0.0 | N/A | N/A | N/A | N/A | N/A | N/A | N/A | N/A | N/A | N/A | 97.2 | 97.6 | 98.6 | 98.9 | 99.1 |
| GBZ-16 | 83.8 | 36.4 | 8.9 | 2.3 | 0.9 | 26.9 | 79.8 | 99.7 | 100.0 | 100.0 | 43.3 | 88.0 | 99.8 | 100.0 | 100.0 | 91.6 | 92.1 | 91.2 | 91.6 | 92.6 |
| GBZ-32 | 82.7 | 35.5 | 9.5 | 3.0 | 1.1 | 26.6 | 83.2 | 99.7 | 100.0 | 100.0 | 45.8 | 90.0 | 99.8 | 100.0 | 100.0 | 90.3 | 91.7 | 91.4 | 92.5 | 92.6 |
| GBZ-64 | 79.6 | 27.4 | 5.6 | 1.5 | 0.5 | 36.1 | 86.3 | 100.0 | 100.0 | 100.0 | 53.5 | 93.1 | 99.9 | 100.0 | 100.0 | 91.3 | 91.4 | 91.9 | 92.2 | 92.7 |
| GBZ-128 | 79.2 | 32.1 | 8.7 | 2.6 | 1.2 | 26.1 | 68.5 | 99.2 | 100.0 | 100.0 | 41.2 | 79.1 | 99.4 | 100.0 | 100.0 | 89.7 | 91.5 | 91.4 | 91.4 | 92.5 |
| GBY-16 | 89.2 | 34.6 | 3.5 | 0.2 | 0.0 | 44.8 | 88.1 | 99.8 | 100.0 | 100.0 | 55.9 | 91.7 | 99.7 | 100.0 | 100.0 | 89.4 | 92.5 | 91.0 | 90.6 | 36.9 |
| GBY-32 | 86.7 | 25.8 | 1.6 | 0.1 | 0.0 | 45.1 | 91.2 | 100.0 | 100.0 | 100.0 | 58.6 | 94.0 | 100.0 | 100.0 | 100.0 | 90.9 | 91.8 | 91.1 | 68.6 | 31.1 |
| GBY-64 | 84.9 | 22.9 | 1.5 | 0.1 | 0.0 | 47.4 | 91.2 | 99.9 | 100.0 | 100.0 | 59.8 | 92.8 | 99.9 | 100.0 | 100.0 | 91.4 | 92.1 | 91.2 | 86.8 | 29.0 |
| GBY-128 | 82.1 | 23.7 | 1.9 | 0.0 | 0.0 | 40.4 | 83.0 | 99.7 | 100.0 | 100.0 | 51.7 | 87.5 | 99.6 | 100.0 | 100.0 | 91.2 | 91.2 | 89.8 | 82.8 | 44.4 |