# OpenReview forum: "Are Generative Classifiers More Robust to Adversarial Attacks?"
_ICLR.cc/2019/Conference_

### Official Review · AnonReviewer2 · 2018-11-02
**Solid and insightful experimental study**

**Rating:** 8
**Confidence:** 3

**Review:**

This paper aims to test the robustness of generative classifiers [1] w.r.t. adversarial examples, considering their use as a potentially more robust alternative to adversarial training of discriminative classifiers. To achieve this, *Deep Bayes*, a generalization of the Naive Bayes classifier using a latent variable model and trained in a fashion similar to variational autoencoders [2] is introduced, and 7 different latent variable models are compared, covering a spectrum of generative or discriminative classification models, with or without bottlenecks. Their DFX and DBX architectures in particular closely match traditional discriminative classifiers, without and with a latent bottleneck.

These 7 models are compared against a large range of adversarial attacks, depending on the kind of noise added (l_2 or l_inf) and how much the adversary can access (the full gradients of the model, its output on training data, or only the model as a black-box). The performance of the models is assessed depending on two criteria: how the performance of the classifier resists to adversarial noise, and how quickly the model can detect adversarial samples. Three methods for detecting adversarial samples are compared: the first (only applicable to generative classifiers) discards samples with a low likelihood, according to the off-manifold assumption [3], the second discards samples for which the classifier has low confidence in its classification (p(y|x) is under some threshold), and the third compares the output probability vector of the classifier on a sample to the mean classification vector of this class over the train data, and discards the sample if the two vectors are too dissimilar (meaning the classifier is over-confident or under-confident).

The main contribution of this paper is the extensive experiments that have been done to compare the models against the various adversarial attacks. While experiments were only done on small datasets like MNIST and CIFAR (generative classifiers don't scale as easily on large image datasets), they nonetheless give very interesting insights and the authors provided encouraging results on applying generative classifiers on features learned by discriminative classifiers. Theirs result shows that generative architecture are in general more robust to the current state-of-the-art adversarial attacks, and detect adversarial examples more easily. The authors also recognize that these results may be biased by the fact that current adversarial attacks have been specifically optimized towards discriminative classifier.

This is a solid paper in my opinion. The experimental setup and motivations are clearly detailed, and the paper was easy to follow. Extensive results and description of the experimental protocol are provided in the appendices, giving me confidence that the results should be reproducible. The results of this paper give interesting insights regarding how to approach robustness to adversarial examples in classification tasks, and provide realistic ways to try and apply generative classifiers in real-worlds tasks, using pre-learned features from discriminative networks.


[1] http://papers.nips.cc/paper/2020-on-discriminative-vs-generative-classifiers-a-comparison-of-logistic-regression-and-naive-bayes.pdf
[2] https://arxiv.org/abs/1312.6114
[3] https://arxiv.org/abs/1801.02774

---

> ### Author Response · Authors · 2018-11-12
> **Thank you for your positive comments!**
>
> Glad to hear that you like our manuscript!
>
> The motivation of this work is to test whether recently developed attacks can break generative classifiers, and our results show that generative classifiers are more robust than discriminative classifiers against recent attacks. Sure research in adversarial attacks and defenses is similar to a "cat and mouse game", and we anticipate in the future an attack tailored to generative classifiers can break our models. But we have done the best to test mainstream attacks available now, and indeed we showed that generative classifiers have properties different from discriminative ones. We expect future work on developing attacks & defenses on generative models can make generative models more powerful and robust!
>
> An important insight of our approach is that the generative classifiers and the proposed detection methods are based on the same generative model. This means detected adversarial inputs are indeed far away from the classifier's manifold, and the classifier's manifold is also an approximation to the data manifold. By contrast, previous approaches require training an extra copy of generative model/auto-encoder. Thus it's very likely that the detector and the classifier do not have aligned manifold representations, thus the classifier cannot enjoy many benefits from the generative model.
>
> We also wanted to encourage future research on combining generative and discriminative approaches. At least from our robustness test on CIFAR-10, this fusion approach indeed is worth further investigations, and it offers an exciting venue for future research.

---

### Official Review · AnonReviewer3 · 2018-11-02
**Interesting direction but some important prior work is missing and evaluation does not yet convincingly support claims**

**Rating:** 4
**Confidence:** 5

**Review:**

In this work the authors propose and analyse generative models as defences against adversarial examples. In addition, three detection methods are introduced and an extension to deep features is suggested.

My main concerns are as follows (details below):
* Important prior work is not mentioned.
* Evaluation with direct attacks is only based on (very few) gradient-based techniques, many results are not reliable.
* There are signs of gradient masking (the common problem of robustness evaluation, in particular of only gradient-based techniques are used).
* The way detection rates are taken into account in the perfect knowledge scenario is confusing.

### Style
I like the idea of testing many different factorisation structures. However, that comes with the drawback that one needs to constantly check back what the abbreviations mean. Together with the three detection methods, the manuscript is quite confusing at times and should definitely be streamlined. One suggestion: remove the detection methods: I did not find any real conclusion about them but they are definitely side-tracking users away from the main results.

### Prior work
There is at least one closely related prior work not mentioned here: the analysis by synthesis model [1]. This model uses a variational auto encoder to learn class conditional distributions and shows high robustness on MNIST. Please make clear what your contribution is over this paper (other than testing several other factorisations).

### Evaluation problems
The robustness of models should be evaluated on different direct attacks ranging from gradient-based to score-based (e.g. NES [2]) to decision-based attacks [3]. Please take a look at [1] to see how a very extensive evaluation might look like. The results can be astonishingly different for different attacks, and so basing conclusion on only one or two attacks is dangerous (in particular if you only use gradient-based ones). One can also see that in your results, just check the variations you get between MIM and PGD. Also, rather then discussing (and showing in detail) results for individual attacks, the minimum adversarial distance for a given sample that can be found by any attack is much more comparable between models (which can also streamline the manuscript).

One can see signs of gradient masking in your results. For example, in Figure 3 the MIM attacks levels out at 20% for the DBX model. That can happen for iterative attacks if the gradient is masked. Similarly, in Figure 5 DBX-ZK (zero knowledge) is better in both accuracy and detection rate than DBX-PKK (which takes the KL-detection method into account and should thus either be better in accuracy or detection rate).

More generally, the perfect knowledge case, in which the attacker knows about the detector, should only count samples as adversarials which evade the detector and change the model decision. Thus, the detection rate should be zero. Otherwise I have no idea what trade-off between accuracy and detection rate you are actually targeting and how to compare the results.

Also, some intermediate results are conflicting with each other. E.g. in 4.1 you state “the usage of bottleneck is beneficial for better robustness”, but for L2 this is not true.

Also, I am not sure how conclusive the grey-box and black-box scenarios really are: since the substitute is basically a DFX or DFZ, it’s unsurprising that adversarials transfer best to those two models.

### Minor
 * In 4.1 you say “as they fail to find near manifold adversarials”, but I don’t see how there can be L-infty adversarials on MNIST that are on-manifold (remember, MNIST pixel values are basically binary). Plus, in the zero-knowledge scenario there is nothing that enforces staying on this manifold.
 * Result presentation (Figure 3/5 & Table 1) is very different for different attack scenarios, which makes them hard to compare. Please unify.
 * Is the L2 distance you report in Table 1 the mean (or median) distance to adversarial examples. If so, GBZ (for which you state that C&W “failed on attacking” has actually a smaller mean adversarial distance than some other models (for which C&W is actually quite successful).
 * Grey-box scenario doesn’t make a lot of sense: since the substitute is basically a DFX or DFZ, it’s unsurprising that adversarials transfer best to those two models. A similar confounder makes the black-box results difficult to interpret.
* Also, taking into account that the paper is two pages longer and thus calls for higher standards

Taken together, I find the general direction of the paper very interesting and I’d definitely encourage the authors to go further. At the current stage, however, I feel that (1) contributions are not sufficiently delineated to prior work, (2) the evaluation is not convincingly supporting the claims and that  (3) the manuscript needs to be streamlined (both in terms of text and figures).

[1] Schott et al. (2018) “Towards the first adversarially robust neural network model on MNIST” (https://arxiv.org/abs/1805.09190)
[2] Ilyas et al. (2018) “Black-box Adversarial Attacks with Limited Queries and Information” ( [https://arxiv.org/abs/1804.08598)](https://arxiv.org/abs/1804.08598))
[3] Brendel et al. (2018) “Decision-Based Adversarial Attacks: Reliable Attacks Against Black-Box Machine Learning Models” (https://arxiv.org/abs/1712.04248)

---

> ### Author Response · Authors · 2018-11-12
> **Our work is concurrent to [1], and our work has significant novel contributions**
>
> Thank you for pointing out the work in [1] to us. Our work is independent to [1], and the majority of the experiments (MNIST & CIFAR binary) was done at the same time when v1 of [1] was released on ArXiv (May 2018). We have submitted evidences to the area chair.
>
> We were not aware of [1] until you kindly pointed it to us. We also found that [1] is currently under review at ICLR 2019 as well, so we feel that this somewhat excuses our lack of knowledge of a paper that can be at best described as concurrent work!
>
> Since our work is concurrent to [1], we believe our evaluation of generative classifiers on MNIST is a novel and significant contribution. **In the revised version we have a paragraph discussing the difference between our work and the concurrent work [1].** We would like to explain the main differences in the following:
>
> 1. We consider a greater number of discriminative and generative classifiers, showing how the factorization of the generative model is important.
>
> 2. We do not only evaluate on MNIST. We look at a CIFAR binary task. This is very important as Carlini and Wagner (2017b) showed that robustness properties shown on MNIST often do not hold on more complicated dataset, e.g. natural images. Our results on CIFAR-binary are consistent with MNIST results, showing that generative classifiers indeed have robustness properties distinct from discriminative ones.
>
> 3. We provide ideas and experiments on how to scale robust generative classifiers to more representative color datasets such as CIFAR-10, by building generative classifiers on pre-trained discriminative features. To the best of our knowledge, this is the first evaluation of the robustness on this fusion model, and we believe our results show that the combined approach offers an exciting research direction.
>
> 4. Our approach gives data log-likelihood meaning to the logits for free, and these logit values can be used for detecting adversarial examples. This is different from previous approaches that require training an extra detector or generative model. Also since the classifier and the detector share the same generative model, detected adversarial inputs are indeed far away from the classifier’s manifold, thus we can use both accuracies and detection rates to verify the “off-manifold” conjecture.
>
> 5. Throughout training and adversarial evaluation we amortize the cost of inference and use K=10 Monte Carlo samples for the latent variable z. By contrast [1]’s method is much much slower: to classify a single image, their method requires K=8000 (!) initial z samples and then 50 gradient steps for z refinement. As these generative classifiers already require more computation over regular CNNs, this computation can often ill be afforded. Also the logit values in [1]’s classifier are ELBOs of log p(x|y) with a very simple Gaussian q distribution, which can be very different from the actual log p(x|y). Instead the logit values of our generative classifier are importance sampling estimates of log p(x|y) which are more accurate.
>
> We thank you for your suggestion on the extra attacks to run. We agree that covering more attacks is always nice! However, we argue that we have already covered a very strong arsenal of attacks. This is evidenced when comparing to [1], as you can see that we evaluate the strongest attacks they find. Furthermore, we have also evaluated the C&W attack, which they have not. We believe this is important as it has been demonstrated to be a very powerful attack (Carlini & Wagner, 2017b), which is also corroborated by our work.
>
> **In the revised version we also show that SPSA as a score based attack fails to fool generative classifiers, which also indicates that the robustness results are unlikely to be caused by gradient masking.**

---

> ### Author Response · Authors · 2018-11-12
> **Apart from independent work statement, on your other questions:**
>
> We address your other concerns in below.
>
> Q1: “perfect knowledge case: definition of adversarial example”
> We show quantitative results for both cases:
> 1. Crafted adv inputs that make the classifier predict wrong class, but are detected by the detection method;
> 2. Crafted adv inputs that both make the classifier predict wrong class, and evade the detection.”.
> The second case might align with your definition of adversarial example in this perfect-knowledge setting, and the value to look for this case is the white space between the top of the shaded bar and 100. However by studying the first case we also show the breakdown between classification tricking & detection avoiding and only classification tricking as an indication of one point on the Pareto front.
>
> Q2: “grey-box/black-box experiment not conclusive”
> What we were trying to do here was allay fears that positive results for the generative models could be due to gradient masking. So we evaluated two types of attack (see the revised manuscript):
> 1. Distillation-based attack. We took a CNN, not vulnerable to the gradient masking problem, and tried to get it to mimic the function learnt by the generative model as closely as possible by training it on its predictions rather than the ground truth predictions. Then we transfer the adversarial examples on the student CNN to the victim classifier.
> 2. Score-based attack. We applied the SPSA attack, which used 2000 samples to numerically estimate the gradient from the victim classifier's outputs.
> If gradient masking is the main reason for the white-box robustness, then the above two attacks should achieve higher success rate (i.e. lower victim accuracies) than the white-box case. This is not the case as shown in the experiments, therefore, gradient masking is unlikely to be responsible for generative classifier's robustness properties.
>
> Q3: “GBZ (for which you state that C&W “failed on attacking” has actually a smaller mean adversarial distance”
> Again our goal is to compare between generative and discriminative classifiers, and we showed in Table 1 that generative classifiers (GFZ, GFY, GBZ, GBY) has higher mean perturbation distances compared to other discriminative classifiers. Also in Figure 4(b) we showed crafted adversarial examples on GFZ (see Figure D.1(h) for GBZ), and many of them are sitting at the perceptual boundary between the original and the adversarial classes. Paper [1] also showed similar results in their Figure 3.
>
> Q4: “usage of bottleneck is beneficial” contradict L_2 results
> We wanted to make this claim for the L_inf case, indeed our quantitative analysis in the appendix was done on L_inf attacks.
>
> Q5: “how there can be L-infty adversarials on MNIST that are on-manifold”
> If an adversarial input has visible noise/artifacts then it’s clearly off the manifold of clean data. But even when the perturbation is difficult for humans to see, generative classifiers are more robust than discriminative ones, see the visualisations in Figure 1 (epsilon=0.1, 0.2) and Figure 8 (epsilon=0.01, 0.02, 0.05).

---

> > ### Comment · AnonReviewer3 · 2018-11-19
> > **Please comment on specific indications of gradient masking**
> >
> > In my initial review I wrote
> >
> > "One can see signs of gradient masking in your results. For example, in Figure 3 the MIM attacks levels out at 20% for the DBX model. That can happen for iterative attacks if the gradient is masked. Similarly, in Figure 5 DBX-ZK (zero knowledge) is better in both accuracy and detection rate than DBX-PKK (which takes the KL-detection method into account and should thus either be better in accuracy or detection rate)."
> >
> > Could you please comment on that?

---

> > > ### Author Response · Authors · 2018-11-20
> > > **On gradient masking issues for DBX, a **discriminative** classifier with bottleneck**
> > >
> > > Thank you for your reply.
> > >
> > > First we emphasize that our goal is to validate that gradient masking is unlikely to explain the robustness of **generative classifiers**. Indeed our experiments provide strong evidence supporting this, and in the revised version we edited the discussions to emphasize this.
> > >
> > > DBX is a **discriminative** classifier. So the following discussion only applies to DBX and does not relate to any observations we had on generative classifiers.
> > >
> > > On MNIST, our observations for FGSM & MIM attacks on DBX are similar to (Alemi et al. 2017). In Alemi et al. (2017), the deep VIB model has the same architecture as DBX in our case, and they showed robustness results on MNIST against FGSM and CW-L2.
> > >
> > > We quickly conducted the white-box MIM attack again but with epsilon = 0.9. This attack achieved 99.5% success rate on MNIST test data, and the success rate for adv inputs crafted on **corrected classified images** is 100%. Furthermore, the white-box PGD can achieve 99.8% success rate on DBX for epsilon = 0.5 (see Figure 3 and table E.2).
> > >
> > > Additionally, our results for CIFAR-binary and CIFAR-10 fusion model show that DBX is less robust than other generative classifiers. E.g. on CIFAR-binary, white-box MIM with epsilon=0.2 achieved 99.9% success rate against DBX (see Table E.7).
> > >
> > > Lastly, our quantitative evaluation on the bottleneck effect (Figure C.4) show that DBX with dim(z)=128 failed to evade white-box MIM attacks. The main text results are for DBX with dim(z)=64, which has stronger bottleneck effect.
> > >
> > > Combining all the above observations, we would conclude that gradient masking is unlikely to exist for DBX. The robustness of DBX against FGSM & MIM with reasonably big epsilon only applies for MNIST, and it requires a carefully selected bottleneck size (or in deep VIB, the beta coefficient that encourages the bottleneck effect).
> > >
> > > From the above conclusion, the robustness & detection results for DBX in the perfect-knowledge case has very little to do with gradient masking.
> > >
> > > We will add a paragraph explaining these in the appendix. Let us know if you have more questions.

---

> > ### Comment · AnonReviewer3 · 2018-11-23
> > **I don't think small tweaks to the manuscript are sufficient to increase the reliability of the results**
> >
> > Let me start out with an important statement: I do like the general idea of your paper and I love that you are probing and testing the space between continuous and generative models. I honestly believe that this can be a great paper. What I am trying to save you from are premature conclusions and a sometimes confusing storyline (in particular with respect to detection rates). Evaluating the robustness of a model to a sufficient degree is really hard. That's the reason why so many proposed defences have later turned out to be false: because the authors did not try hard enough. You have performed many different experiments in total, and I absolutely applaud your efforts on the overall analysis. The big problem I see is that in each individual experiment you are not trying hard enough to make sure that discriminative and generative models are evaluated sufficiently well to draw reliable conclusions.
> >
> > Let me go again through my main points, taking into account the revisions you have made.
> >
> > 1) It makes little sense to discuss results of individual attacks because the efficiency of attacks varies naturally between models. Concentrate on the minimum adversarial distance across instead (this will simplify your main text and the figures).
> >
> > 2) The black-box distillation attack is prone to be biased towards discriminative models because the substitute itself is a discriminative model.
> >
> > 3) It’s great that you added SPSA, but it’s suspicious that this attack is so much worse than PGD. At least in L2 black-box attacks with numerical gradient estimates or decision-based attacks tend to work as well or better than gradient-based attacks on both discriminative and generative networks.
> >
> > 4) It’s still strange and a potential sign of gradient masking that MIM plateaus on DBX. I don’t see the same effect in (Alemi et al. 2017).
> >
> > 5) In Figure 5 DBX-ZK (zero knowledge) is better in both accuracy and detection rate than DBX-PKK (which takes the KL-detection method into account and should thus either be better in accuracy or detection rate). I raised this in my original review already.
> >
> > 5) The manuscript is really confusing because you are looking at so many independent dimensions: different datasets (MNIST, binary CIFAR, CIFAR), different networks (generative, discriminative), different attack types (gradient-based, distillation, score-based), different settings (perfect knowledge, zero knowledge) and different metrics (L2, Linfinity). In addition, you are discussing results exclusive to one setting instead of focusing on those things that generalise across many.
> >
> > The manuscript needs to be seriously streamlined and I’ve mentioned some things that might help, in particular: perform many attacks and primarily report the minimum across all attacks as a single metric that you discuss (and which is much more reliable than each individual metric), and take out the whole detection part (or tell me which of the main conclusions is based on it).
> >
> > 6) “usage of bottleneck is beneficial” contradict L_2 results. You seem to agree, so for what reasons are you still report this? What conclusions should the reader take from this?
> >
> > Given all the issues I pointed out, I still feel that this work is not yet ready for publication. I do, however, very much encourage the authors to finish the full analysis and to attack both discriminative and generative models as best as you possibly can.

---

> > > ### Author Response · Authors · 2018-11-25
> > > **We believe our conclusion is correct; detection results are very important; no gradient masking for generative classifiers (1/2)**
> > >
> > > Thank you for comments and also for your complimentary remarks as to the general idea of our paper and overall analysis. We believe the conclusion on the robustness of generative classifiers is correct, in fact our findings on MNIST agrees with the observations of Schott et al. (2018) which is concurrent to us.
> > >
> > > We quickly run through these points below, some of them are further clarified in the latest revision. Please let us know whether our replies have addressed your concerns.
> > >
> > > 1. “Focus on minimum perturbation distance only”
> > > In the latest revision we added the minimum perturbation distance results for L_inf attacks. Still, our visualisation gives further information indicating the change in success relative to the change in perturbation size, which when coupled with adversarial example visualizations, indicate the robustness of the model across a range of perturbations, from imperceptible to perceptible.
> > >
> > > 2. “Remove all detection results”
> > > We strongly disagree: detection is another important contribution of our work. Our work is the 1st to make the classifier and the detection mechanisms share the **same generative model**, i.e. the classification and detection decisions use the **same model manifold** (as a proxy to data manifold). This means, in order to fool both the classifier and the detection, the adversary needs to find “blind spots” near the manifold, which is unlikely to exist if the generative model approximates the data distribution well. Indeed, in our results, we clearly see that as the perturbation size increases, the robustness of the models naturally decreases but the detection rates increase.
> > >
> > > Our approach is different from previous approaches that require a separate training of the generative model/auto-encoder, and there is no guarantee that an independently trained generative model has its manifold aligned with the classifier’s manifold. Therefore, previous results on bypassing detection methods (Carlini & Wagner, 2017b, Athayle et al., 2018) do not extend to our case, and our detection tests evaluated the effectiveness of the proposed detection methods.
> > >
> > > 3. “Gradient masking, and black box is biased towards discriminative models”
> > > The main goal of this experiment is to try to provide evidence against gradient masking being able to explain the generative classifiers’ good results. If gradient masking was present then we would expect to be able to find adversarial examples more easily in the grey/black box cases than in the white box setting.
> > >
> > > It is for this end that we use a CNN as the substitute model. We know that CNNs are powerful function approximators, and so hopefully they can model the classification boundaries of our generative classifiers well (indeed on MNIST, the substitute achieved >99% agreements with the victim), whilst also providing useful gradients to attackers. We do not use generative classifiers for the surrogate models, as this would not help us rule out the generative classifiers’ model structure being susceptible to gradient masking.
> > >
> > > The aim of this section is *not* to compare the transferability of examples between discriminative and generative classifiers, which is investigated later in the “cross-model attack transferability” section.
> > >
> > > (see below for continued reply)

---

> > > > ### Author Response · Authors · 2018-11-25
> > > > **We believe our conclusion is correct; detection results are very important; no gradient masking for generative classifiers (2/2)**
> > > >
> > > > (continuing the above comments)
> > > >
> > > > 4. “Suspicious SPSA results”
> > > > We disagree with your claim that “black-box attacks with numerical gradient estimates tend to work as well or better than gradient-based attacks”. In fact this statement contradicts with empirical results in Schott et al. (2018): in Table 1 of that paper (L_2 and L_inf cases), for CNN the FG(S)M/BIM/MIM attacks with exact gradients work significantly better than the same attacks but with score-based (-GE) gradients. This can also be seen in the Brendel et al. (2017) as well, where against cleanly trained networks, the median L_2 distortion was higher for decision-based attacks compared to strong gradient-based attacks (CW). Importantly, if a victim classifier does not suffer from gradient masking (which is the case for CNN), then the gradient should provide useful information, and gradient-based attacks with exact gradients should perform better than those with numerically estimated gradients.
> > > >
> > > > We have provided extensive evidence in the distillation-based attack experiments that the trained generative classifiers are unlikely to have gradient masking issues. Therefore, the fact that SPSA (gradient estimation) has a lower success rate than PGD (gradient-based) simply suggests again that the generative classifiers do not suffer from gradient masking.
> > > >
> > > > For your info, we've tried our best to make SPSA a strong black-box attack: we use almost the same hyper-parameters as in (Uesato et al., 2018), except that the number of ES samples is M=2000 due to the limit of the GPU. Initial testing also showed that M=1000 gives similar results as M=2000.
> > > >
> > > > Also note that our models are trained on clean data only, so the results in Schott et al. (2018) on adversarial training (“Madry et al.”), indicating that gradient estimation yields the same success rate as gradient-based with lower minimum adversarial distance,  is unrelated to our sanity check experiments, as the tested model was adversarially trained explicitly to defend against gradient-based adversarial attacks.
> > > > Further empirical evidence that adversarially trained models suffer from gradient masking is provided in Papernot et al. (https://arxiv.org/pdf/1602.02697.pdf Table 4) and discussed in the Cleverhans blog (http://www.cleverhans.io/security/privacy/ml/2017/02/15/why-attacking-machine-learning-is-easier-than-defending-it.html).
> > > >
> > > > 5. On DBX results and the bottleneck effect:
> > > > Alemi et al. (2016) tested FGSM on MNIST, showing that by increasing beta (i.e. having bigger bottleneck effect), the robustness results also improves (see Fig 5 in their paper). As MIM is basically “iterative” FGSM with momentum updates, it is unsurprising that DBX is more resistant than other discriminative classifiers against white-box MIM. In the appendix we have a detailed study on the bottleneck effect, and we showed that if dim(z) is too big (e.g. dim(z)=128), then the bottleneck effect is not significant, and in this case MIM achieved 99.8% success rate against DBX at eps = 0.3. Please also refer to the previous reply in below for details.
> > > >
> > > > 6. Perfect-knowledge experiments on DBX:
> > > > We compared DBX-ZK to DBX-PK0 and DBX-PKK.
> > > > a) Recall that the PK0 attack sets lambda=0 and only considers the randomness in the sampling procedure of DBX. Therefore the fact that PK0 results are close to ZK results suggests that randomness has little effect on the efficiency of the PGD attack.
> > > > b) PKK optimises a different loss function (eq 4), and it is possible that PGD gets stuck at a bad local minimum. We leave the investigation of better optimisation methods to future work. Also in the detection plots of Figure 3, KL-detection achieves ~100% detection rates for PGD attacks against DBX, meaning that PGD tends to find adversarial examples that fool DBX with **too low or too high** confidence. Therefore, because PKK wants to evade KL-detection, it is reasonable that PKK achieves lower fooling rates (i.e. higher victim accuracy) than ZK and PK0.
> > > >
> > > > 7. “Why reporting L_2 CW results”
> > > > CW-L_2 is a representative white-box attack that we need to test. We do not intend to claim that the bottleneck effect extends to L_2 attacks. The results we showed here are: (i) generative classifiers are in general more robust than discriminative ones; (ii) the required perturbation for generative classifiers are higher, and (iii) the perturbed images on generative classifiers are ambiguous to humans as well. The third result also agrees with Figure 3 in Schott et al. (2018).

---

> > > > > ### Comment · AnonReviewer3 · 2018-11-26
> > > > > **please take the time for a major revision of the manuscript**
> > > > >
> > > > > I understand that you are trying to address my comments and concerns under the impression of a tight rebuttal deadline which leaves little room for more than cosmetic changes in the manuscript. However, addressing the shortcomings with respect to robustness evaluation and presentation of results that I pointed out in two detailed reviews will require a major revision of the manuscript and the experiments. This did not yet happen and so I unfortunately do not see room to increase my score.

---

### Official Review · AnonReviewer1 · 2018-11-14
**Good work with thorough experimental study**

**Rating:** 6
**Confidence:** 3

**Review:**

This work investigates an interesting direction of improving robustness of classifiers against adversarial attacks by using generative models. The authors propose the *deep Bayes classifier*, which is a deep LVM based extension of naive Bayes. Furthermore, the authors extensively explore 7 possible factorisations of the classifier. Thorough experiments are conducted to assess the capability of defending or detecting adversarial examples.  Besides, the authors incorporate discriminative features to generative classifiers and demonstrate clear robustness gain.

### Highlights
* This work proposes an attractive direction -- the use of generative model in defending or detecting adversarial attacks. I suggest this idea should be follower by more further studies.
* The presented models are quite straight-forward but exhibit good robustness against attacks listed in the experiments.
* Various structural possibilities of the graphical model are examined which is preferable and helps assess the effectiveness of generative classifiers.

### Minors
* Although the major point here is robustness against adversarial attacks, as mentioned by the authors, the performance on clear cases (i.e. no attacks) is unsatisfactory. Also, experiments on CIFAR are too much simplified (only 2 very unlike classes) and therefore not very convincing.
* For the combination of generative classifier and discriminative features, I’m curious about the results on the clear CIFAR-10 multi-class problem. It should be a very positive plus if results are satisfactory.
* The writing is sometimes hard to follow. For examples, many ad-hoc abbreviations are used across the paper causing difficulties of understanding the core idea and results.

### Conclusion
In general, this paper brings our attention to a previously less investigated but seemingly promising research direction, i.e. robustness of generative model against adversarial attacks. The idea is insightful and proposed models are straight-forward. While only on small-scale  problems (with the presence of attacks), extensive experimental results in this paper can assist further study on this field. Thus, I recommend this paper to be accepted.

---

> ### Author Response · Authors · 2018-11-15
> **We did test the full CIFAR-10 multi-class case**
>
> Thank you for your review, we are glad that you liked our paper in general. We thank for your suggestions on better writing and will include them in revision.
>
> We clarify on our CIFAR experiments in below.
>
> 1. The test on CIFAR-binary is an important contribution:
> 1a) Carlini & Wagner (2017b) have shown that many defense techniques that works on MNIST didn’t work very well on natural images. Therefore it is important to have a natural image classification task, to see whether the robustness results on MNIST also extends to natural images.
> 1b) We did try full generative classifiers on CIFAR-10 multi-class classification. Unfortunately it is still a research challenge to make full generative classifiers work beyond MNIST. In fact we have tried even more powerful architectures like PixelCNN++ to parameterise each of p(x|y), and the clean accuracy in this case is 72.4%. Therefore we don’t think it’s fair to compare this PixelCNN++ based classifier against e.g. VGG-16 (clean accuracy >93%), since the gap on clean accuracy is huge.
> 1c) Still due to the importance of testing natural image classification tasks, we derived from CIFAR-10 a binary classification task that the deep Bayes classifiers work reasonably well (>90% accuracy), and tested the robustness of them on this task. The general conclusions here are consistent with MNIST experiments, providing evidences that the robustness results of generative classifiers do extend to natural images.
>
> 2. The results of the fusion model are indeed on**full CIFAR-10 multi-class classification**.
> 2a) Since full generative classifiers don’t work very well on full CIFAR-10, we decided to take discriminative features from VGG-16 and train generative classifiers on the features. We have the clean accuracy results reported in table D.2 for multi-class classification. In this case all classifiers have >88% accuracy on clean data, so we can have a reasonably fair comparison here.
> 2b) The robustness tests (Figure 10) still favours the fusion model, and the bottleneck discriminative classifiers (DBX-) didn’t improve robustness against PGD & MIM. To the best of our knowledge this is the first robustness test on this fusion model. Our results show that combining generative modelling and discriminative features is an exciting future research direction.
> 2c) Importantly, we show that using lower-layer features for the fusion model (i.e.~the model is less “discriminative”) returns even better results. This is a clear evidence favouring generative classifiers, and we expect in future developments, a full generative classifier that can achieve >90% clean accuracy on CIFAR-10 will be even more robust.
>
> Thank you for your review again, let us know if you have more questions.

---

### Official Review · AnonReviewer4 · 2018-12-14
**Interesting work but confusing presentation and some missing areas**

**Rating:** 4
**Confidence:** 4

**Review:**

This paper explores the potential of generative models for adversarial robustness. It presents some interesting and well formulated findings but is lacking in some meaningful ways.

-It does a good job of summarizing the literature though it misses out on some very recent but relevant work
-It introduces three detection methods and extensively evaluates them  against several zero-knowledge attack types. Notably all the attack types are gradient based methods.
-They present detailed performance metrics on the zero-knowledge attack, and introduce the perfect knowledge attack but do not present equivalently detailed results.
-The results are  presented in a confusing manor, and the paper is perhaps done a disservice by the inclusion of a very large number of tables both in the main text and the appendix without the necessary writing required to situate the reader.
-The paper also briefly mentions the ongoing debate around "off-manifold" conjecture, only to them assume its correctness and based the entire work on the premise.

###Highlights
*well written introduction and a good overview of generative modeling
*very good visualizations and explanation of the detection methods
*thorough results on zero-knowledge gradient based attack detection rates
*interesting direction on adversarial examples

### Areas for improvement
*more time should be spent on discussing the off manifold conjecture; the paper is based on the correctness of this conjecture though recent work has called its validity into question
*only gradient based attacks are discussed though the paper title suggests robustness to all attacks
*perfect-knowledge attacks should be explored in much more detail
* the paper should streamline the results and findings, the large number of tables and results will confuse the reader and do not add to the argument
*recent work on generative models for adversarial robustness should be discussed (https://arxiv.org/abs/1805.09190, https://arxiv.org/abs/1811.06969)
*the two class cifar10 results are not particularly convincing as this is a substantial simplification of the cifar10 problem
* the results on the full cifar10 data set make use a the extracted features of a pre-trained discriminative cifar10 network, this introduces the problem at layer activation adversarial attacks, but this is not mentioned in the paper (https://arxiv.org/pdf/1511.05122.pdf)

Overall i think the work shows promise but is not yet ready for acceptance

---

> ### Author Response · Authors · 2018-12-15
> **Our work is concurrent to Schott et al. (2018); many of the concerns have been addressed in the manuscript**
>
> Thank you for your comments.
>
> Short answers to your the major concerns:
>
> 1. Our work is highly novel, the two "prior work" you mentioned are after or concurrent to our work. We have discussed this in section 5.
> 2. We did tested score-based attack (SPSA) which is **gradient-free**. Results show that the robustness of generative classifiers is NOT due to gradient masking.
> 3. We did not pre-assume the correctness of the "off-manifold" conjecture, in fact our experiments serve as empirical verification of this conjecture.
> 4. The CIFAR-10 binary experiments are the best we can do to evaluate **fully generative classifiers** on **natural images**.
> 5. Observations on layer activation attacks against the **fusion model** cannot directly transfer to robustness results of **fully generative classifiers**. We have discussed similar issues in section 5.
>
> Detailed reply on your major questions:
>
> 1. "two previous work not discussed"
> 1a) Our work is at least concurrent to Schott et al. (2018) (in fact, earlier, we have submitted evidence to the AC).
> 1b) Schott et al. (2018) is also under submission to ICLR 2019.
> 1c) Frosst et al. (2018) was online on **Nov 16 2018**, which is 1.5 months after ICLR submission deadline, and we were not aware of it.
> 1d) We have discussed Schott et al. (2018) in revision (see the second paragraph in section 5).
> 1e) In the below "summary of the Q&A" comment we extensively discussed the main difference of our work to Schott et al. (2018), and we consider the difference to be very significant.
>
> 2. "Notably all the attack types are gradient based methods"
> We did tested the SPSA attack (Uesato et al. 2018) which is based on evolutionary strategies and thus **gradient-free**.
> 2a) This attack has been shown to be very successful against **gradient masking based defences**.
> 2b) Our results show that SPSA worked **worse** than white-box attacks on generative classifiers.
> 2c) Therefore, we conclude that the robustness of generative classifiers is NOT due to gradient masking.
>
> 3. "off-manifold conjecture pre-assumed"
> We did not pre-assume the correctness of the "off-manifold" conjecture:
> 3a) We were interested in verifying the "off-manifold" conjecture for both generative and discriminative classifiers. If for small epsilon the classifiers have high error, then this conjecture might not be true in practice.
> 3b) The detection methods depend on the correctness of the "off-manifold" conjecture. Therefore, if empirically the detection methods don't work even when we have a very good generative model (which is the case for MNIST), then, this conjecture might not be true in practice.
> 3c) Our empirical results clearly show that (i) generative classifiers are more robust than discriminative ones (especially when epsilon is small); (ii) the marginal/logit detection methods did work.
> 3d) Observing the above, we are confident to say we have empirically verified the "off-manifold" conjecture on generative classifiers.
>
> 4. "two class cifar10 results are not particularly convincing"
> We presented this example because:
> 4a) Carlini & Wagner (2017b) stated that MNIST results might not transfer to natural images. Therefore we wanted to see if our findings on MNIST is still true on natural images. By contrast Schott et al. (2018) only studied MNIST.
> 4b) We did try full CIFAR-10, but none of the existing generative models (VAE/GAN/flow-based/autoregressive models) can achieve satisfactory **clean accuracy** on it.
> 4c) Therefore we instead studied this two-class CIFAR-10 problem in order to evaluate **fully generative classifiers**. We then presented the fusion model for full CIFAR-10 to address the scalability issue.
>
> 5. "the problem at layer activation adversarial attacks"
> Thanks for pointing the relevant work which we shall cite. However:
> 5a) The fusion model is not **fully generative**, and it contains **discriminative features** which might be vulnerable to attacks.
> 5b) Therefore failures of the fusion model against layer activation attacks do not imply the failure of **fully generative classifiers**.
> 5c) To be clear, we have no intension to oversell our results and claim generative classifiers are robust to **all attacks**. In fact we spent a full paragraph to discuss this issue, please see the third paragraph in section 5.

---

### Author Response · Authors · 2018-11-19
**Revision available, please consider updating your review**

Thank you for your reviews, we have revised our submission according to some of your suggests.

Summary of major edits:
1. We reported the SPSA attack (a black-box score-based attack) results on the full generative classifiers. Results show that gradient masking is unlikely to explain the robustness of the generative classifiers.
2. We reported minimum L_inf perturbations on the L_inf white-box attacks (mean/median reported). Again generative classifiers require larger perturbations to be fooled.
3. We discussed the differences between our work and (Schott et. al. 2018) which is **independent and concurrent to us**. Our approach is much much more scalable, we tested on both MNIST and CIFAR-10, and we showed that the graphical model of the LVM does matter for robustness.

Again we believe our contribution is highly novel and significant. Please consider the revised version, and it would be much appreciated if you can update your reviews accordingly.

Best,
Paper356 Authors

---

### Author Response · Authors · 2018-11-25
**Revision v2 available, please consider updating your review**

Dear reviewers:

We’ve made another update to the manuscript. Summary of new edits are as follows:

1. In the end of section 3, we add a paragraph to discuss the novelty and importance of the proposed detection methods. Our paper is the 1st to make the generative classifier and the detection work together, as they share the **same** generative model. Our results show that existing attacks fail to fool both the classifier and the detection method at the same time.
2. We revised the grey-/black-box distillation experiments, focusing on showing that the generative classifiers are unlikely to suffer from gradient masking. In this section we revised the figures, and added a table that directly compares the accuracy and minimum perturbation of white-/grey-/black-box attacks.

We also repeat the edits in revision v1 (as compared to the Sept submission):
3. We reported the SPSA attack (a black-box score-based attack) results on the full generative classifiers. Results show that gradient masking is unlikely to explain the robustness of the generative classifiers.
4. We reported minimum L_inf perturbations on the L_inf white-box attacks (mean/median reported). Again generative classifiers require larger perturbations to be fooled.
5. We discussed the differences between our work and (Schott et. al. 2018) which is **independent and concurrent to us**. Our approach is much much more scalable, we tested on both MNIST and CIFAR-10, and we showed that the graphical model of the LVM does matter for robustness.

We strongly believe our contribution is highly novel, timely and significant. Please consider the latest revised version, and it would be much appreciated if you can update your reviews accordingly.

Best,
Paper356 Authors

---

### Author Response · Authors · 2018-11-27
**Summary of the Q & A**

We are grateful to all the reviewers for taking the time to review our paper. We’ve looked carefully at all the questions, and made revisions to answer the feedback as best as possible. Here’s a summary of the Q & A.

Q1: “prior work (Schott et al., 2018) not mentioned”
A1: Our work is concurrent with Schott et al., (2018). We cited this paper in revision and discussed the major differences:
1a) We performed extensive evaluation on both MNIST and CIFAR-10. Schott et al., (2018) only tested their method on MNIST and reached to similar conclusions as ours.
1b) We investigated different graphical models for both generative and discriminative classifiers. Schott et al., (2018) only tested one architecture.
1c) We amortized the cost of inference with a recognition network, and the importance sampling only needs K=10 latent variable samples. By contrast, Schott et al., (2018)’s model requires K=5000 samples and 50 optimisation steps to predict y for a single x.
1d) We proposed new detection methods utilizing the learned model and evaluated their effectiveness in experiments. Schott et al., (2018) did not discuss this.
1e) We proposed the generative-discriminative fusion model as a scalable solution to big data and big networks. Schott et al., (2018) did not discuss this.

Q2: “experiments on CIFAR are small scale”
A2: We did have large-scale tests on full CIFAR-10 multi-class classification, and the baseline there was a VGG-16 network. The generative-discriminative fusion model is more robust than VGG-16 and other discriminative classifiers, showing that the robustness properties of generative classifiers do extend to CIFAR-10.

Q3: “possible gradient masking issues”
A3: In addition to the existing attacks against surrogate models, we added the SPSA attack (a state-of-the-art score-based attack) in revision. We tried our best to make SPSA strong, by using M=2000 ES samples to estimate a single gradient, and it took 3 days on a Tesla P100 GPU to perform attacks against a generative classifier on 1,000 clean MNIST inputs. Both the distillation-based attacks and the SPSA attacks worked worse than the white-box attacks, suggesting that gradient masking is not the explanation of the robustness of the tested generative classifiers.

Q4: “focus on minimum perturbation distance only”
A4: In revision we added visualisations and tables on the min distortion of the L_inf attacks. We believe other metrics are worth presenting as well, see answers to questions below.

Q5: “take out the whole detection part”
A5: Our detection method is the first to make the classifier and detector use the **same generative model manifold** (as a proxy to the data manifold). This is distinct from all previous attempts, which train a **separate** generative model and thus have no guarantee to have an aligned manifold representation. Therefore previous success on attacking detection methods (Carlini & Wagner 2017, Athayle et al., 2018) do not directly transfer to our proposal, and would need to be reassessed. Our results clearly show that as the perturbation size increases, the robustness of the models naturally decreases but the detection rates increase as well. This means it is hard to fool both the classifier and the detection simultaneously.

Q6: “issues on DBX and the bottleneck effect”
A6: We first emphasize that DBX is still a discriminative classifier.
DBX is unlikely to suffer from gradient masking: white-box MIM with eps=0.9 achieved 100% success rate against DBX on MNIST; white-box PGD with eps=0.3 achieved 94% success rate against DBX on MNIST; DBX failed against white-box attacks on CIFAR-binary and CIFAR-10. Our quantitative study in the appendix also shows that by increasing dim(z), reducing the bottleneck effect, DBX becomes less robust against FGSM & MIM on MNIST.

---

### Author Response · Authors · 2019-01-07
**Thank you for the AC & reviewers, and a latest summary of the Q & A**

We would like to thank the AC and the reviewers to engage in discussions during the feedback time. Our presentation has room for improvement and again thank you for the suggestions on this.
For reference, we summarise the major questions raised by the reviewers, and our answers to them.


Q1: “prior work (Schott et al., 2018) not mentioned”
A1: We thank the reviewers for pointing out Schott et al., (2018) which is concurrent. Our work is concurrent with Schott et al., (2018). The first version was submitted to arXiv in Feb 2018, and the previous version was presented at ICML 2018 TADGM workshop (oral) in July 2018.

We cited this paper in revision and discussed the major differences:
1a) We performed extensive evaluation on both MNIST and CIFAR-10. Schott et al., (2018) only tested their method on MNIST and reached to similar conclusions as ours.
1b) We investigated different graphical models for both generative and discriminative classifiers. Schott et al., (2018) only tested one architecture.
1c) We amortized the cost of inference with a recognition network, and the importance sampling only needs K=10 latent variable samples. By contrast, Schott et al., (2018)’s model requires K=5000 samples and 50 optimisation steps to predict y for a single x.
1d) We proposed new detection methods utilizing the learned model and evaluated their effectiveness in experiments. Schott et al., (2018) did not discuss this.
1e) We proposed the generative-discriminative fusion model as a scalable solution to big data and big networks. Schott et al., (2018) did not discuss this.

Q2: “focus on minimum perturbation distance only, and take out the whole detection part”
A2: In revision we added visualisations and tables on the min distortion of the L_inf attacks. However:
Our detection method is the first to make the classifier and detector use the **same generative model manifold** (as a proxy to the data manifold). This is distinct from all previous attempts, which train a **separate** generative model and thus have no guarantee to have an aligned manifold representation. Therefore previous success on attacking detection methods (Carlini & Wagner 2017, Athayle et al., 2018) do not directly transfer to our proposal, and would need to be reassessed. Our results clearly show that as the perturbation size increases, the robustness of the models naturally decreases but the detection rates increase as well. This means it is hard to fool both the classifier and the detection simultaneously.


Q3: "off-manifold conjecture pre-assumed"
A3: We did not pre-assume the correctness of the "off-manifold" conjecture:
3a) We were interested in verifying the "off-manifold" conjecture for both generative and discriminative classifiers. If for small epsilon the classifiers have high error, then this conjecture might not be true in practice.
3b) The detection methods depend on the correctness of the "off-manifold" conjecture. Therefore, if empirically the detection methods don't work even when we have a very good generative model (which is the case for MNIST), then, this conjecture might not be true in practice.
3c) Our empirical results clearly show that (i) generative classifiers are more robust than discriminative ones (especially when epsilon is small); (ii) the marginal/logit detection methods did work.
3d) Observing the above, we are confident to say we have empirically verified the "off-manifold" conjecture on generative classifiers.


Q4. "two class CIFAR-10 results are not particularly convincing"
A4: We presented the CIFAR-binary example because:
4a) Carlini & Wagner (2017b) stated that MNIST results might not transfer to natural images. Therefore we wanted to see if our findings on MNIST is still true on natural images.
4b) We did try full CIFAR-10, but none of the existing generative models (VAE/GAN/flow-based/autoregressive models) can achieve satisfactory **clean accuracy** on it. Therefore we instead studied this two-class CIFAR-10 problem in order to evaluate **fully generative classifiers**
4c) We then presented the fusion model for full CIFAR-10 to address the scalability issue. The baseline there was a VGG-16 network. The generative-discriminative fusion model is more robust than VGG-16 and other discriminative classifiers, showing that the robustness properties of generative classifiers do extend to CIFAR-10.

Q5: “possible gradient masking issues”
A5: We added the SPSA attack (a state-of-the-art score-based attack by Uesato et al. 2018) in revision. We tried our best to make SPSA strong, by using M=2000 ES samples to estimate a single gradient, and it took 3 days on a Tesla P100 GPU to perform attacks against a generative classifier on 1,000 clean MNIST inputs. Both the distillation-based attacks and the SPSA attacks worked worse than the white-box attacks, suggesting that gradient masking is not the explanation of the robustness of the tested generative classifiers.

---

### Meta-Review · Area_Chair1 · 2018-12-14
**Good approach to adversarial robustness, but the reviewers have doubts. A more thorough, streamlined experimental setup would help.**

**Confidence:** 3
**Recommendation:** Reject

**Metareview:**

Adversarial defense is a tricky subject, and the authors are to be commended for their novel approach to this problem. The reviewers all agree that there is promise in this approach. However, after reviewing the discussion, they have all come to the conclusion that the robustness of your generative model needs to be more thoroughly explored. Regarding gradient masking, there are other attacks like a manifold attack that use gradients that can be explored as well. Regarding SPSA, it would be helpful perhaps to also include other numerical gradient attacks to ensure that SPSA is stronger and working as intended.

Essentially, the reviewers would all like to see a more streamlined version of this paper that removes any doubt about the efficacy of the generative approach. Once that is established, additional properties and features can be explored.

Also note that for the purposes of these reviews and discussion, Schott et al. was considered as concurrent work and not prior work.